# Experimental Study on Energy Dissipation Performance and Failure Mode of Web-Connected Replaceable Energy Dissipation Link

**Zhanzhong Yin [1,2,\*], Zhaosheng Huang [1,\*], Hui Zhang [1] and Dazhe Feng [1]**

[1]   School of Civil Engineering, Lanzhou University of Technology, Lanzhou 730050, China
[2]   Western Engineering Research Center of Disaster Mitigation in Civil Engineering of Ministry of Education, Lanzhou University of Technology, Lanzhou 730050, China
\*   Correspondence: yinzhanzhong@lut.cn (Z.Y.); zshenghuang@163.com (Z.H.)

**Abstract:** In the current design method of the eccentrically braced frame structure, the energy dissipation link and the frame beam are both designed as a whole. It is difficult to accurately assess the degree of damage through this method, and it is also hard to repair or replace the energy dissipation link after strong seismic events. Meanwhile, the overall design approach will increase the project's overall cost. In order to solve the above mentioned shortcomings, the energy dissipation link is designed as an independent component, which is separated from the frame beam. In this paper, the energy dissipation link is bolted to the web of the frame beam. Both finite element simulation and test study of eight groups of energy dissipation links have been completed to study their mechanical behaviors, and the energy dissipation links have been studied in the aspects of length, cross section, and stiffener spacing. The mechanical behaviors include the energy dissipation behavior, bearing capacity, stiffness, and plastic rotation angle. The results indicate clearly that the hysteretic loop of links in the test and finite element analysis is relatively full. By comparing the experimental and finite element simulation data, it can be found that the general shape and trend of hysteretic loop, skeleton curve, and stiffness degradation curve are basically the same. The experiment data explicitly shows that the energy dissipation link of web-connected displays good ductility and stable energy dissipation ability. In addition, the replaceable links possess good rotational capacity when the minimum rotation angle of each specimen in the test is 0.16 rad. The results of the experiment show that the energy dissipation capacity of the link is mainly related to the section size and the stiffening rib spacing of the link. The energy dissipation ability and deformation ability of the link is poorer as the section size becomes larger; meanwhile, these abilities are reduced with the decrease of the stiffening spacing. The experiment result shows that the damage and excessive inelastic deformations are concentrated in the link to avoid any issues for the rest of the surrounding elements, and the links can be easily and inexpensively replaced after strong seismic events. The results are thought provoking, as they provide a theoretical basis for the further study of the eccentrically braced frame structure with replaceable links of web-connected. In future work, the author aims to carry out his studies through optimized design methodology based on the yielding criterion.

**Keywords:** replaceable links; finite element analysis; experimental studies; energy dissipation; hysteretic behavior; ductility; eccentrically braced frame

---

## 1. Introduction

In the current practice, the energy dissipation link and frame beam are designed as a whole, which makes the replacement procedure rather difficult. Replaceable links have been proposed in recent years

in order to avoid this problem [1–3]. There are some researches have studied replaceable steel links in steel-concrete hybrid structures. AlessandroZona presented a hybrid coupled wall (HCW) system that was made of a single reinforced concrete (RC) wall coupled to two steel side columns by means of steel links. The design objective is to reduce or possibly avoid the damage in the RC wall while concentrating the seismic damage on the replaceable steel links [4]. Alper studied a benchmark building frame with and without bolted dissipative beam splices [5]. In Morelli's paper, the development, calibration, and experimental validation of two component-based models of dissipative steel links connecting a reinforced concrete wall to a steel gravity frame is presented [6]. Replaceable steel coupling beams (RSCB) [7] have been proposed as an alternative to conventional reinforced concrete (RC) coupling beams for the enhanced seismic resiliency of coupled wall systems. In these new frame types, damage mainly concentrates on the bolted dissipative beam splices (links) acting as "structural fuses", which can be easily and inexpensively replaced after strong seismic events.

　　Replaceable links are isolated from adjacent structural members to absorb deformation and energy in seismic events [8,9], so as to protect the lateral load resisting systems. Replaceable link is extensively used in EBFs, as it can be used to evaluate the post-earthquake construction performance quickly and accurately and it can be easily repaired and replaced [10–14].

　　The length of the link is a very important parameter for the strength, stiffness, and ductility of the EBFs system [15–18]. The link length ratio $\rho = e/(M_p/V_p)$, where e is the link length, $M_p$ is the plastic moment capacity, and $V_p$ is the plastic shear capacity of the link, is usually used to represent the yielding behavior of a link. According to the relative relation between the link's plastic moment and shear ability, the link length can be divided into short links, where the shear yielding of the web is found to be the dominant factor and long links where the flexural yielding controls the link performance [19,20]. However, an intermediate link would experience a combination of both shear and flexural yielding. Past studies by Popov and his colleagues [21–26] has shown that the strength and ductility of short link are better than long link and they suggested the value of short link length ratio for shear behavior control, $\rho < 1.6$, which is still in use in many design specifications [27–32]. Stratan and Dubina [2] conducted the first study that specifically targeted EBFs with horizontal links that can be easily removed and replaced after an earthquake. The author drew a conclusion that link length ratio ($\rho$) should be limited to 0.8 then the link would have proper cycle behavior through studying a kind of link that connects beam with the bolt end plate. Further numerical and experimental studies of Dubina [33] confirmed the applicability of the concept and presented a general methodology for designing structures with removable dissipative elements and two applications. Okazaki et al. [34] conducted cyclic loading tests on connection specimens of 23 full-scale eccentric support frames. The connection width ratio, the connection super strength coefficient, and the effect of loading process on the connection were tested. There are mainly two types of connection between energy dissipation beam section and frame beam in k-type eccentrically braced frames, which are end plate connection type and web connection type, respectively. Since the 2000s, Dubina and Stratan [33,35,36] studied a link-to-beam connection with bolted end-plates and concluded that the link length ratio should be limited to 0.8 for the sake of preferable hysteretic behavior. Bolted end-plate connections for replaceable shear links were fabricated by using complete joint penetration welds [36–38], partial joint penetration welds [39], and fillet welds [40,41]. In 2011, Mansour [13] conducted an experimental study on two link designs: a w-type endplate connection with bolted connections and a back-to-back c-type connection with bolted or welded connections. The links exhibited a very good ductile behavior, developing stable and repeatable yielding. Bozkurt [42] conducted further experimental studies on replaceable links for steel eccentrically braced frames (EBFs) in 2017. The author put forward a replaceable link detail, which was based on splicing the directly connected braces and the beam outside the link. The validity of this method has been verified by eight near-full-size EBF tests under quasi-static cyclic loading. In addition, Stephens and Yin [39,43] recommended a bolted end plate connection detail for replaceable links that utilized fillet welds at the link-to-endplate interface recently.

In past researches, there were many on the end plate connection type energy dissipation beam, but few on the web connection type. Based on the previous researches [42–45], it can be found that the failure of the end plate connection type energy dissipation link is mainly due to the failure of the flange and the failure of the weld, which reduces the constraint of the flange on the web and further leads to the reduction of the bearing capacity of the link. This paper presents an experimental research program that utilizes web connections at the link-to-beam interface. A total of eight cyclic quasi-static full-scale cyclic tests are performed to study their inelastic seismic performance. Based on the results of the experimental and numerical study, the links of web-connected exhibit a very good ductile behavior, stable energy dissipation capacity, and high rotation capacities, which is also easy to replace. The relation between the length of the link and the energy dissipation coefficient are analyzed. The influence of geometric parameters, such as length ratio and stiffener spacing on the seismic performance of the link web-connected, are obtained by experimental results analysis. Finally, some useful conclusions are drawn to provide suggestions and design recommendations for the follow-up researches.

## 2. Experimental Program

In this experiment, the pseudo-static test method was used to apply the horizontal shear force to the web-connected replaceable links of eight different parameters, and the mechanical properties of the independent replaceable energy dissipation link with web-connected were obtained.

### 2.1. Specimen Design and Material Properties

By referring to the design idea that was proposed by Nabil [46–48], the energy dissipation link and the frame beam of the eccentrically braced frame connected pass through the high-strength bolts. The web of double channel steel of link was connected with the web of frame beam of EBFs. Stiffening ribs were set on both sides of double channel steel. Q235B steel was used for channel steel and stiffening ribs, and the bolts were high-strength steel bolts of 10.9 M20. Weld hole was set on stiffening rib, so as to avoid the effect of stress concentration on test results in order to avoid weld stress concentration between channel steel and stiffening rib. Through theoretical research and numerical simulation, it can be known that the main parameters affecting the mechanical properties of energy-dissipated links include the length of the link, the size of the section of the link, the spacing of the stiffeners, the high thickness ratio of the web(h/tf), and the width ratio of the flange(b/2tw) [10,21,48]. Eight groups of replacement energy dissipation links web-connected have been selected for testing through finite element analysis and comprehensive comparison of the above parameters. Table 1 shows details parameters of the test specimens.

**Table 1.** Parameter table of specimens.

| Serial Number | Section | Length e/mm | ρ | Number of Stiffening Ribs | Stiffener Spacing | Yield Type |
|---|---|---|---|---|---|---|
| RSL-1 | [160 × 63 × 6 × 8 | 212 | 1.16 | 2 | 10@92 | shear |
| RSL-2 | [160 × 63 × 6 × 8 | 322 | 1.76 | 3 | 10@96 | flexural |
| RSL-3 | [160 × 63 × 6 × 8 | 322 | 1.76 | 2 | 10@202 | flexural |
| RSL-4 | [200 × 73 × 8 × 10 | 322 | 1.57 | 3 | 10@96 | shear |
| RSL-5 | [200 × 73 × 8 × 10 | 322 | 1.57 | 3 | 10@202 | shear |
| RSL-6 | [200 × 73 × 8 × 10 | 412 | 2.01 | 4 | 10@91 | flexural |
| RSL-7 | [220 × 77 × 8 × 10 | 322 | 1.47 | 3 | 10@96 | shear |
| RSL-8 | [250 × 80 × 8 × 12 | 322 | 1.21 | 3 | 10@96 | shear |

Detailed drawings of specimens are shown in Figure 1, which only shows details regarding specimens of RSL-4. Details of other specimens are basically the same.

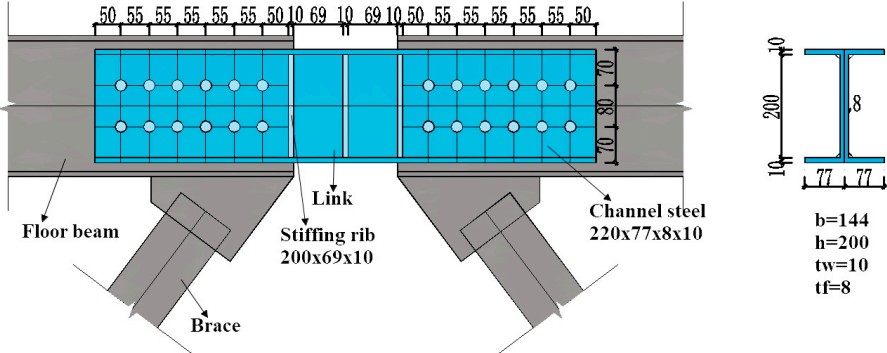

**Figure 1.** Detail drawing of test specimen.

## 2.2. Material Properties

The section of the link was composed of two symmetrical channel steel that was made of welded steel plate. Before the formal experiment, the steel plate of the same batch as the test specimen was taken for the material property test. According to relevant regulations, the yield strength and ultimate strength of specimens of various specifications were determined according to the test results. Figure 2 shows materials testing bars. Test data of steel properties are shown in Table 2.

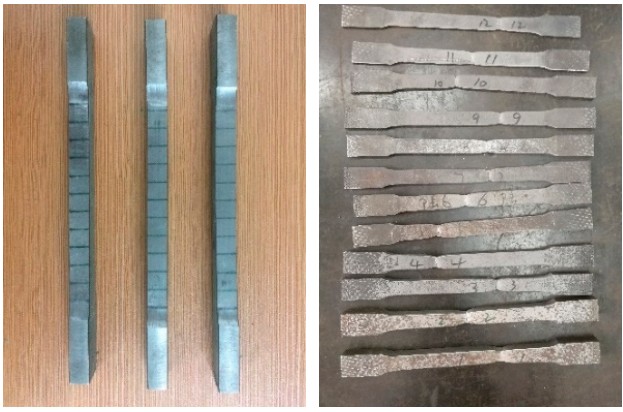

**Figure 2.** Test chart of steel properties.

**Table 2.** Test data of steel properties.

| Thickness (mm) | Measured Value (mm) | Yield Strength fy/MPa | Ultimate Strength fu/MPa | MOE E/MPa | Elongation δ/% |
|---|---|---|---|---|---|
| 6 | 5.53 | 285 | 425 | 212,570 | 30.66 |
| 8 | 7.49 | 290 | 430 | 209,870 | 24.96 |
| 10 | 9.60 | 270 | 420 | 197,100 | 28.20 |
| 14 | 13.54 | 283 | 423 | 199,561 | 30.06 |

## 2.3. Test Setup and Instrumentation

Figure 3a shows the test setup. In order to ensure that the replaceable link can accept the pure shear force, the author adjusted the loading center of the hydraulic jack, the energy dissipation center, and the center of the dumpling stand on the right side to make them on a straight line. The loading device can alter the length of link by adjusting the position of high strength screw rod and dumpling holder pin. The web of double channel steel was connected with the web of frame beam through high-strength bolt, so multiple replacements of the link can be realized.

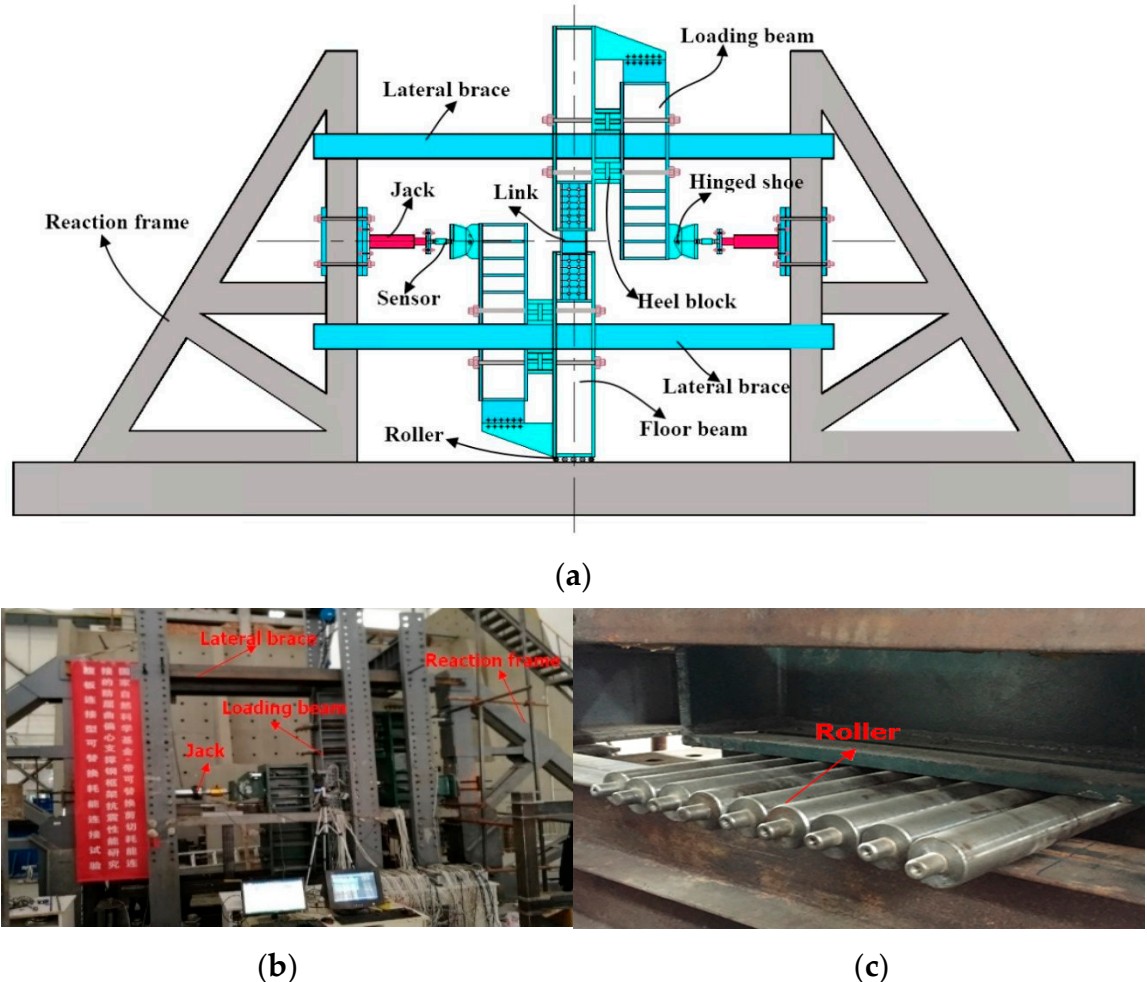

**Figure 3.** Experimental setup. (**a**) Details of test setup; (**b**) Photo; and, (**c**) Roller.

The loading frames that measure 500 mm × 600 mm × 40 mm × 30 mm were arranged on one side of the frame beams measuring 500 mm × 300 mm × 12 mm × 30 mm, respectively. Besides the connections of the four high-strength tie rods between the loading beam and the frame beam, both of their ends were connected to the end plates measuring 588 mm × 300 mm × 24 mm through a row of seven 10.9-class M26 high-strength bolts, and some box pads were placed between them. In addition, a pressure sensor was pinned to the jack at one end and pinned to the left hinge support at the other hand. They were arranged between the left reaction frame and the left loading frame. The right hinge support was pinned to the right loading frame at one hand and it was pinned to the right reaction frame at the other hand. Figure 3b shows the loading site. A row of rollers that can horizontally slide was added at the bottom of the lower frame beam to ensure the sliding of the bottom frame beam during the test (see Figure 3c). At the same time, the rollers were smeared with lubricant to reduce friction [49].

### 2.4. Test Point Layout and Data Collection

The same finite element model was established by Abaqus software, and the stress distribution on the specimen can be obtained by numerical simulation. Subsequently, the strain gauge was pasted on the specimen, which can not only obtain the stress distribution on the energy-dissipated connection, but also optimize the layout of the strain gauge.

Stick strain gauge and strain rosette were arranged on the back side of the web of the channel steel [49]. A strain gauge was pasted inside the web stiffened ribs. Figure 4 shows the arrangement of strain gauge of each test specimen.

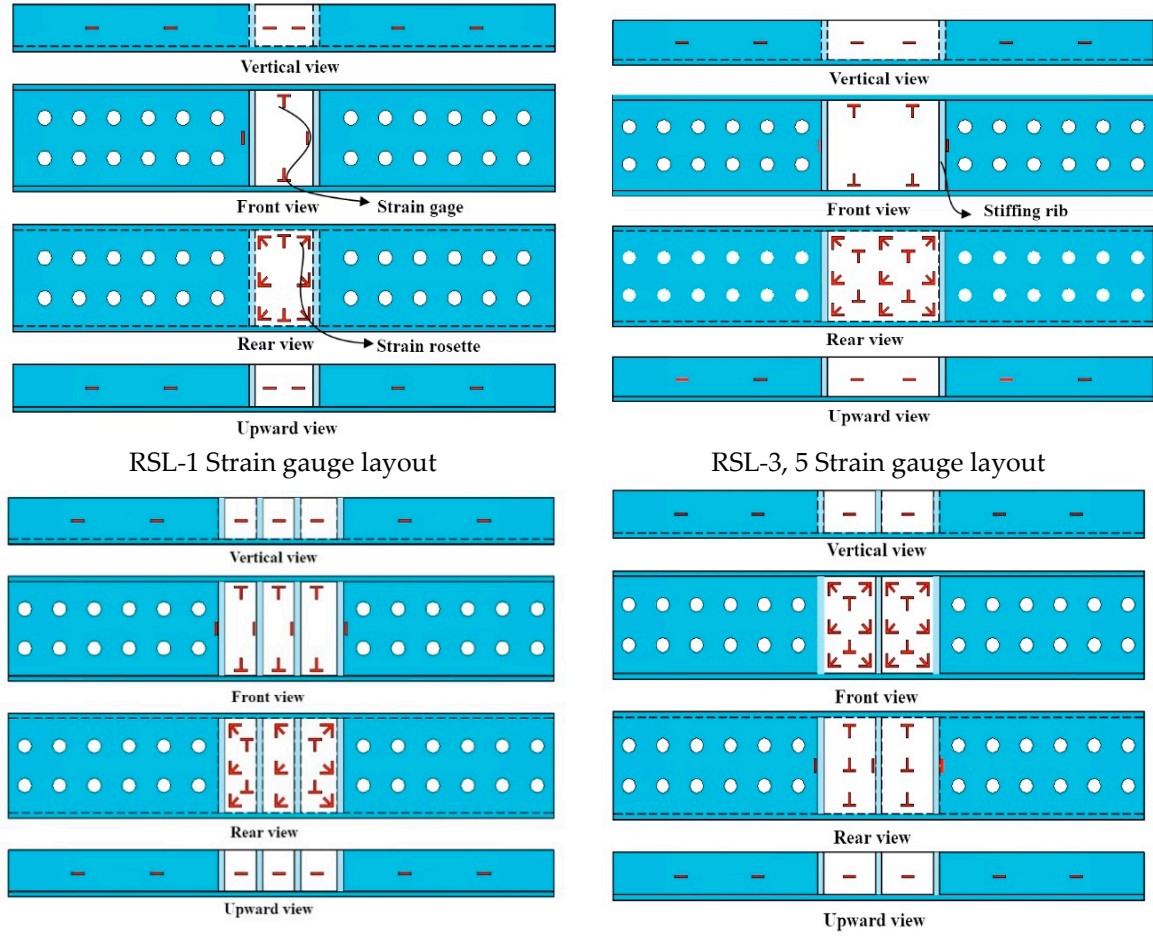

**Figure 4.** The measurement arrangement of strain gauge and strain rosette.

All data in the test are collected through the DH3816 static strain acquisition system and DH5922N dynamic signal acquisition and analysis system. The static strain acquisition system of DH3816 mainly collects the displacement and strain of specimens. The dynamic signal test and analysis system of DH5922N was used to collect the dynamic data in the test.

*2.5. Loading Protocol*

In this test, the displacement loading method was adopted to the applied horizontal cyclic load for the specimen and the constant amplitude loading was implemented. The amplitude of displacement loading was integer times of yield displacement. As $\pm\Delta y/4$, $\pm\Delta y/2$, $\pm3\Delta y/4$, $\pm\Delta y$, $\pm2\Delta y$, $\pm3\Delta y$, $\pm4\Delta y$, $\pm5\Delta y$, $\pm6\Delta y$, $\pm7\Delta y$, etc. $\Delta y$ is the yield displacement obtained by numerical simulation of specimens. Figure 5 shows the loading protocol.

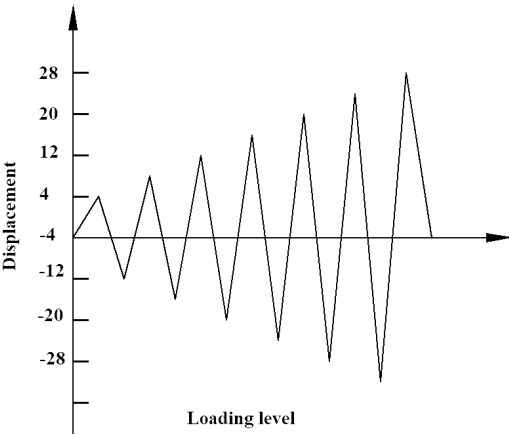

**Figure 5.** Loading Protocol.

## 3. Experimental Results and Discussion

The cyclic load is applied to the specimen until the specimen is damaged by the pulling and pressing jack. For convenience, the channel steel on one side of the line displacement meter is collectively referred to as channel steel A, and the other side is called channel steel B; the link is divided into three parts, the upper and lower connection parts, and the intermediate energy dissipation part.

### 3.1. Failure Mode and Damage Processes

At the beginning of the experiment, the specimen RSL-1 is basically in the elastic range with no obvious change. With the increase of displacement, the specimen enters the plastic stage from the elastic stage. When the displacement reaches 24 mm, varnished leather bulging appears outside the right flange of channel B steel (as shown in Figure 6a). When the displacement reaches −30 mm, the varnished leather of the inner panel of the stiffened ribs in the area of link and the bolt position next to the stiffened ribs on the upper side will be bulging (as shown in Figure 6b). As the displacement reaches 30 mm, the lateral flange on the front side of channel A steel shows the bending phenomenon of stiffening ribs to the inside (as shown in Figure 6c). When it was in the second cycle of 30 mm, the lacquer skin of the flange of channel A is bulging, and that of the channel steel of B side is bulging, and the peeling area of lacquer skin is further expanded (in Figure 6d). The overall deformation of link continues to increase as the displacement continues to increase (as shown in Figure 6e). When the displacement reaches −60 mm, the varnished leather of the web plate of the upper bolt position of the stiffening rib of channel B steel falls off completely (as shown in Figure 6f). When the displacement reaches 72 mm, the web is torn and the specimen is damaged (as shown in Figure 6g).

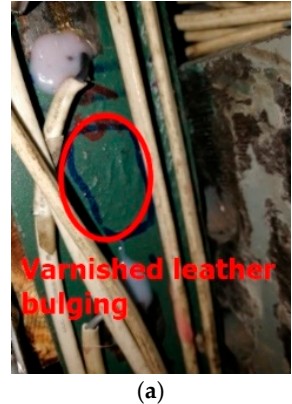 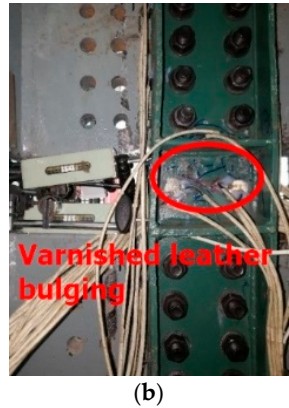 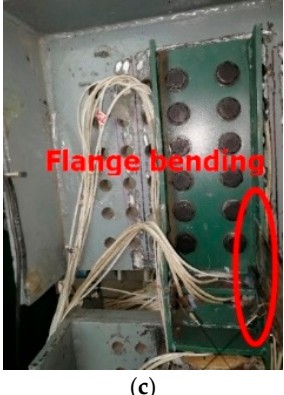

(**a**)    (**b**)    (**c**)

**Figure 6.** *Cont.*

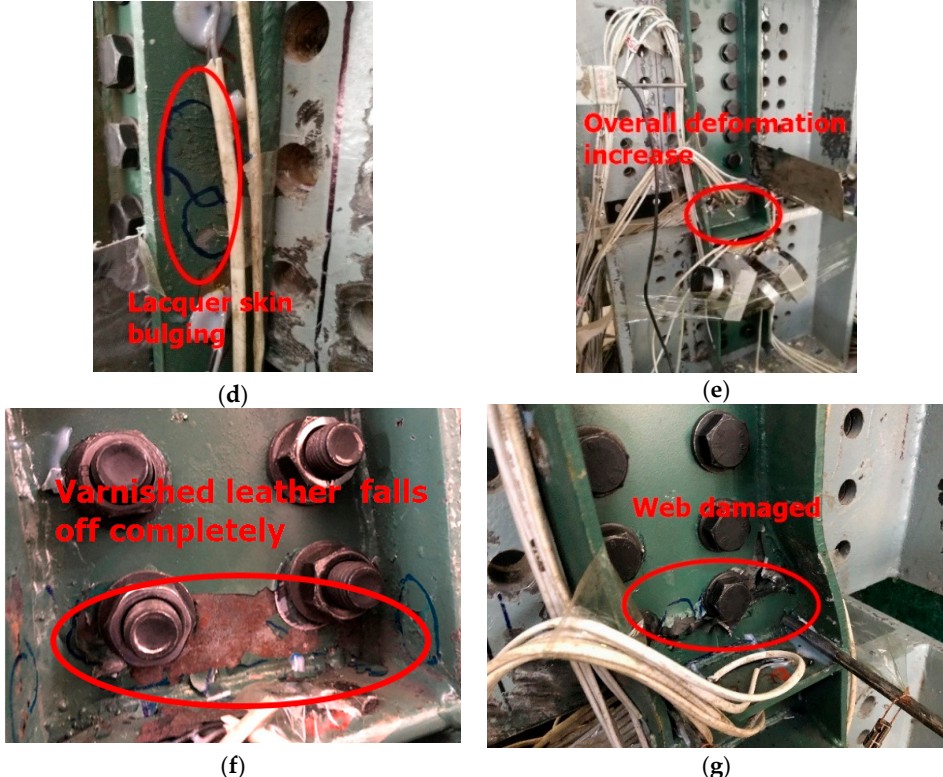

**Figure 6.** Failure modes of RSL-1. (**a**) vanished leather bulging; (**b**) vanished leather bulging; (**c**) flange bending; (**d**) lacquer skin bulging; (**e**) overall deformation increase; (**f**) vanished leather falls off; (**g**) web damaged.

There are similar experimental phenomena between specimen RSL-2 and specimen RSL-3. When the displacement reaches −15 mm, the bolt is under stress and noise accompanied the course of bulging. The central flange of channel B steel is slightly bent outwards (as shown in Figure 7a). When the displacement reaches −25 mm, both channel A and channel B show flange bending deformation, and the lower side flange of the link is curved inward (as shown in Figure 7b). When the displacement reaches 40 mm, the lacquer skin on the inner side of the right flange on the lower side of the channel B steel cracked and bulged (as shown in Figure 7c). When the displacement reaches −55 mm, the lacquer skin of web plate of channel A and B steel bulged. As the displacement reaches −74 mm, the load will no longer continue to increase and the experiment stopped (as shown in Figure 7d).

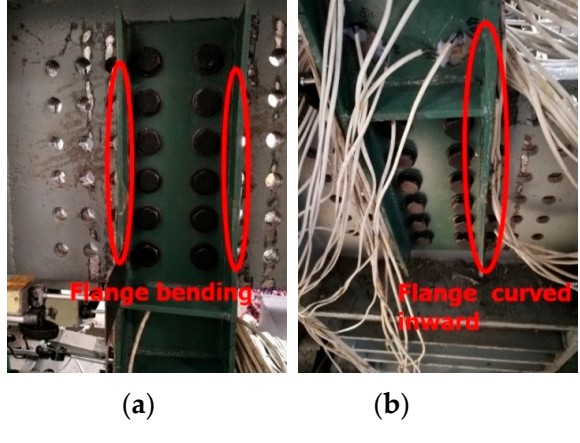

**Figure 7.** *Cont.*

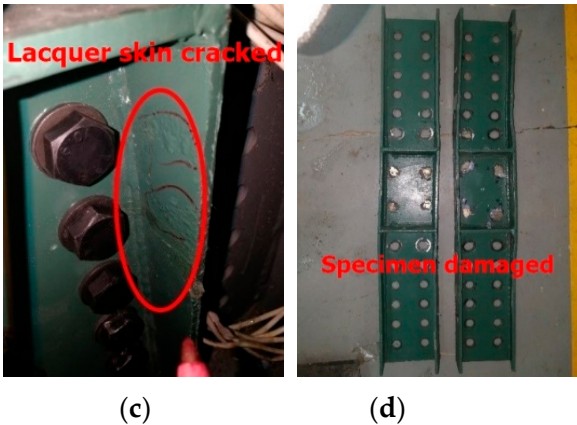

(c)          (d)

**Figure 7.** Failure modes of RSL-2. (**a**) flange bending; (**b**) flange cured inward; (**c**) lacquer skin cracked; (**d**) specimen damaged.

As previously mentioned, both channel A and channel B steel of specimen RLS-4 show flange bending deformation, and the upper lateral flange of the connection part is all curved inward as the displacement reaches −28 mm (as shown in Figure 8a). When the displacement reaches 28 mm, varnished leather bulging appears outside the right flange of the channel A steel (as shown in Figure 8b). When the displacement reaches 34 mm, the bending deformation of upper and lower flanks at the joint of channel steel of A and B surface continues to increase (as shown in Figure 8c,d). When the displacement reaches −55 mm, the painted skin of web plate of channel A and B steel bulged. When the displacement reaches −72 mm, the load will not continue to increase until the test stops.

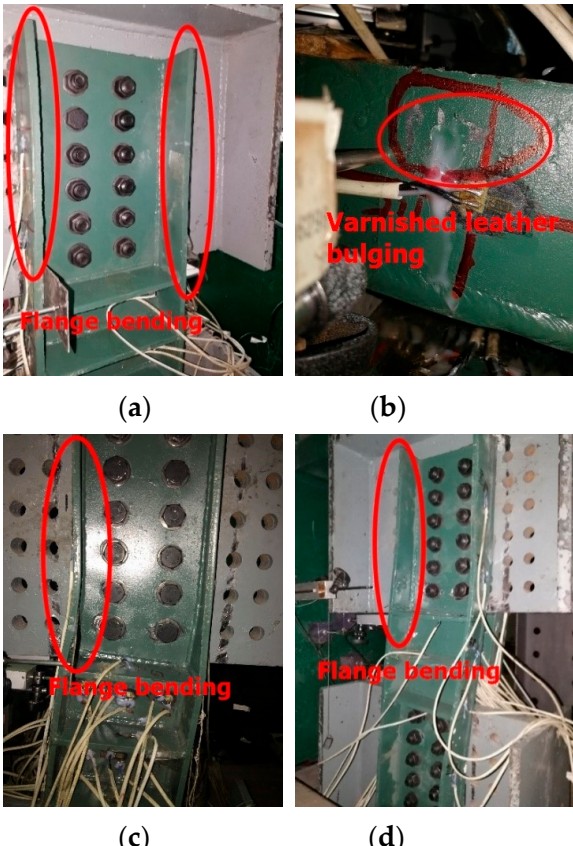

(c)          (d)

**Figure 8.** Failure modes of RSL-4. (**a**) flange bending; (**b**) vanished leather bulging; (**c**) flange bending; (**d**) flange bending.

Samples RLS-5 to RLS-8 have similar experimental phenomena as what mentioned before, which are not described here one by one.

Through the tests and a large number of finite element analyses, it can be known that, under cyclic loading, the deformation of the specimens is concentrated on the replaceable links, and most of the stress reaches the ultimate strength of the materials. However, the rest of the structure (frame column, frame beam, brace) remains elastic. The residual deformation of the frame is very small. According to the follow-up test data we did (Table 3), after plastic damage has occurred on the link, the residual deformation of the frame is very small. The maximum stress on frame beam, column, and brace is very small, which is far less than the yield stress of the material. The above analysis shows that the frame beams columns and brace are still in the elastic stage after several energy dissipation links replacement. The link has good replaceable properties. The high-strength bolt is removed from the original frame by an electric wrench when the link is broken. It is feasible to replace the link after adjusting to the original frame position with the jack, as shown in Figure 9.

**Table 3.** The specimen plastic corner.

| specimen | RSL-1s | RSL-2s | RSL-3s | RSL-4s |
| --- | --- | --- | --- | --- |
| plastic corner | 0.230 | 0.170 | 0.140 | 0.080 |
| specimen | RSL-5s | RSL-6s | RSL-7s | RSL-8s |
| plastic corner | 0.100 | 0.150 | 0.060 | 0.050 |

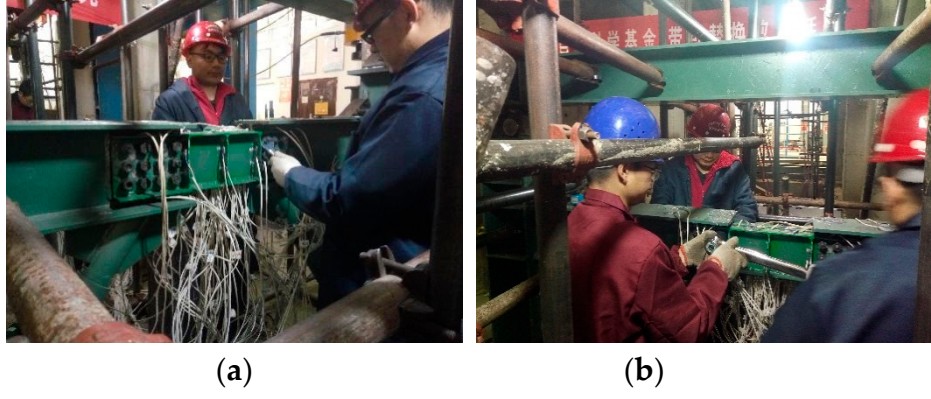

(**a**)  (**b**)

**Figure 9.** Replacement of energy-dissipating links. (**a**) replacement of connection; (**b**) replacement of connection.

From the above-mentioned damage processes, eight specimens showed primarily two types of failure modes. The failure mode of Specimens L2, L3, and L6 was shear failure, the flange stress was mainly distributed on the flange between the first row and the third row bolts. The deformation was mainly concentrated on the web and flange near the first row of bolt holes that were connected with the frame beam. The bolt holes on the web were shear deformed and the flange was bent. Specimens L1, L4, and L5, L7, L8 failed due to local buckling of the flange after shear yield of the web, which was likely to be caused by the stress at the web point when it reached the limit stress yield before that at the flange point. In addition, the local buckling load of web and the stability of the connection can be improved by setting transverse stiffener. In this test, a total of eight groups of replacement links with web connection types have been tested, and a total of eight links have been replaced. According to the results of the experiment, the link of the web plate connection type is convenient for the post-earthquake repair and replacement, which can achieve the purpose of post-earthquake repair and substitute and reduce the overall cost of the project at the same time.

### 3.2. Stress Analysis

According to Equations (1)–(5), the data that were collected by DH3816 are processed to obtain the variation of stress value of test specimen during the test.

$$\left.\begin{array}{c}\varepsilon_1\\\varepsilon_2\end{array}\right\} = \frac{\varepsilon_{0°} + \varepsilon_{90°}}{2} \pm \frac{1}{2}\sqrt{\left(\varepsilon_{0°} + \varepsilon_{45°}\right)^2 + \left(\varepsilon_{45°} + \varepsilon_{90°}\right)^2} \tag{1}$$

$$\sigma_1 = \frac{E}{1 - \mu^2}(\varepsilon_1 + \mu\varepsilon_2) \tag{2}$$

$$\sigma_2 = \frac{E}{1 - \mu^2}(\varepsilon_2 + \mu\varepsilon_1) \tag{3}$$

$$\gamma_{xy} = 2\varepsilon_{45°} - \varepsilon_{0°} - \varepsilon_{90°} \tag{4}$$

$$\tau_{xy} = \frac{\sigma_1 - \sigma_2}{2} \tag{5}$$

where $\varepsilon_{0°}$, $\varepsilon_{45°}$, and $\varepsilon_{90°}$ are, respectively, the strain in the horizontal, oblique, and vertical directions of the strain rosette, and $\mu$ is the Poisson ratio of material. The serial number of the flange and web strain gauge of the RSL-1 specimen is shown in Figure 10. In this paper, the stress of each measuring point when the shear stress of the web enters the yield and the stress of each measuring point in the limit state are selected for analysis.

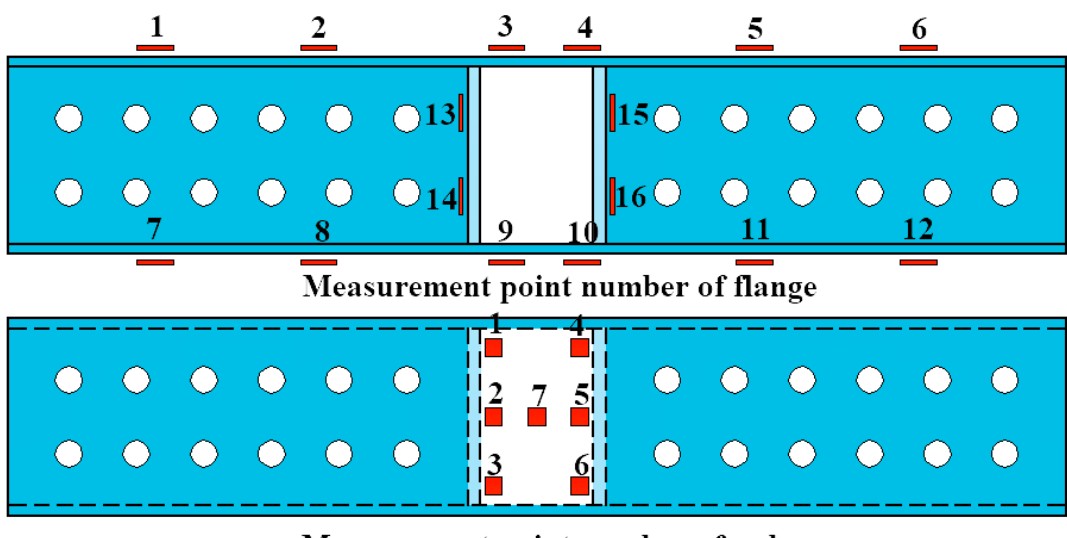

Measurement point number of flange

Measurement point number of web

**Figure 10.** Measurement point number of RSL-1.

The flange and web strain gauge numbers of RSL-1 are shown in Figure 10. Through detailed analysis of the stress data of RSL-1 (Tables 4 and 5), the shear stress of the diagonal line 1 and 6 points of the link web board reach 162.43 Mpa and 169.79 Mpa, respectively. However, the stress at other points is in the elastic stage, which indicated that part of the web begins to shear yield. With the increase of displacement, the shear yield area of the link web gradually developed from the No. 1 point to the No. 2 and 3 points, and from the No. 6 point to the No. 5 and No. 4 point, until the whole web shear yield. The shear stress at points 1, 2, 3, 4, 5, and 6 on the web reaches the shear yield stress, while the flange is still in the elastic stage. It can be seen that the link web was first shear yield. With the increase of shear stress, the value of normal stress gradually increases. It can be seen from Table 4 that both normal stress and shear stress on the web reach the ultimate strength of the material.

**Table 4.** The stress value of RSL-1 when D = −18 mm (0.0847 rad).

| Position | No. | Normal Stress (Mpa) | No. | Normal Stress (Mpa) |
|---|---|---|---|---|
| Flange | 1 | 26.05 | 9 | −0.29 |
| | 2 | 140.16 | 10 | −94.31 |
| | 3 | 101.98 | 11 | −78.56 |
| | 4 | 69.07 | 12 | −92.90 |
| | 5 | −109.66 | 13 | 108.65 |
| | 6 | −161.77 | 14 | 137.93 |
| | 7 | −85.43 | 15 | 111.48 |
| | 8 | −11.12 | 16 | 17.77 |
| Stiffening rib | 17 | −27.67 | 18 | −28.13 |
| | 19 | −36.96 | 20 | −35.75 |

| Position | No. | Normal stress (Mpa) | Normal stress (Mpa) | Shear stress (Mpa) |
|---|---|---|---|---|
| Web | 1 | 144.11 | −180.75 | 162.43 |
| | 2 | 133.66 | −118.65 | 126.16 |
| | 3 | 182.82 | 75.11 | 53.85 |
| | 4 | 154.04 | −109.90 | 131.97 |
| | 5 | 93.14 | −91.12 | 92.13 |
| | 6 | 72.49 | −147.20 | 169.79 |
| | 7 | −19.99 | −12.12 | - |

**Table 5.** The stress value of RSL-1 when D = −72 mm (0.3274 rad).

| Position | No. | Normal Stress (Mpa) | No. | Normal Stress (Mpa) |
|---|---|---|---|---|
| Flange | 1 | 29.69 | 9 | −16.16 |
| | 2 | 407.15 | 10 | −107.64 |
| | 3 | ≥425 | 11 | −57.15 |
| | 4 | ≥425 | 12 | ≤−425 |
| | 5 | ≤-425 | 13 | ≥425 |
| | 6 | ≤-425 | 14 | ≥425 |
| | 7 | −164.60 | 15 | 352.22 |
| | 8 | 58.57 | 16 | 61.59 |
| Stiffening rib | 17 | ≤−425 | 19 | ≤−425 |
| | 18 | ≤−425 | 20 | ≤−425 |

| Position | No. | Normal Stress (Mpa) | Normal Stress (Mpa) | Shear Stress (Mpa) |
|---|---|---|---|---|
| Web | 1 | ≥425 | ≥425 | ≥245 |
| | 2 | ≥425 | ≥425 | ≥245 |
| Web | 3 | ≥425 | ≥425 | ≥245 |
| | 4 | ≥425 | ≥425 | ≥245 |
| | 5 | ≥425 | ≥425 | ≥245 |
| | 6 | ≥425 | ≥425 | ≥245 |
| | 7 | ≥425 | −184.79 | - |

The similar rule can be obtained through detailed analysis of the test data of the other specimens: when the large area of the energy-dissipated web has entered the limit state, and while the flange is still in the elastic stage, it can be observed that the force form of the link is the first shear yield of the web.

### 3.3. Hysteretic Performance

By applying low cycle reciprocating load to the replacement energy dissipation link of the web plate connection type, the load displacement curve of the link under cyclic load is obtained, as shown in Figure 11.

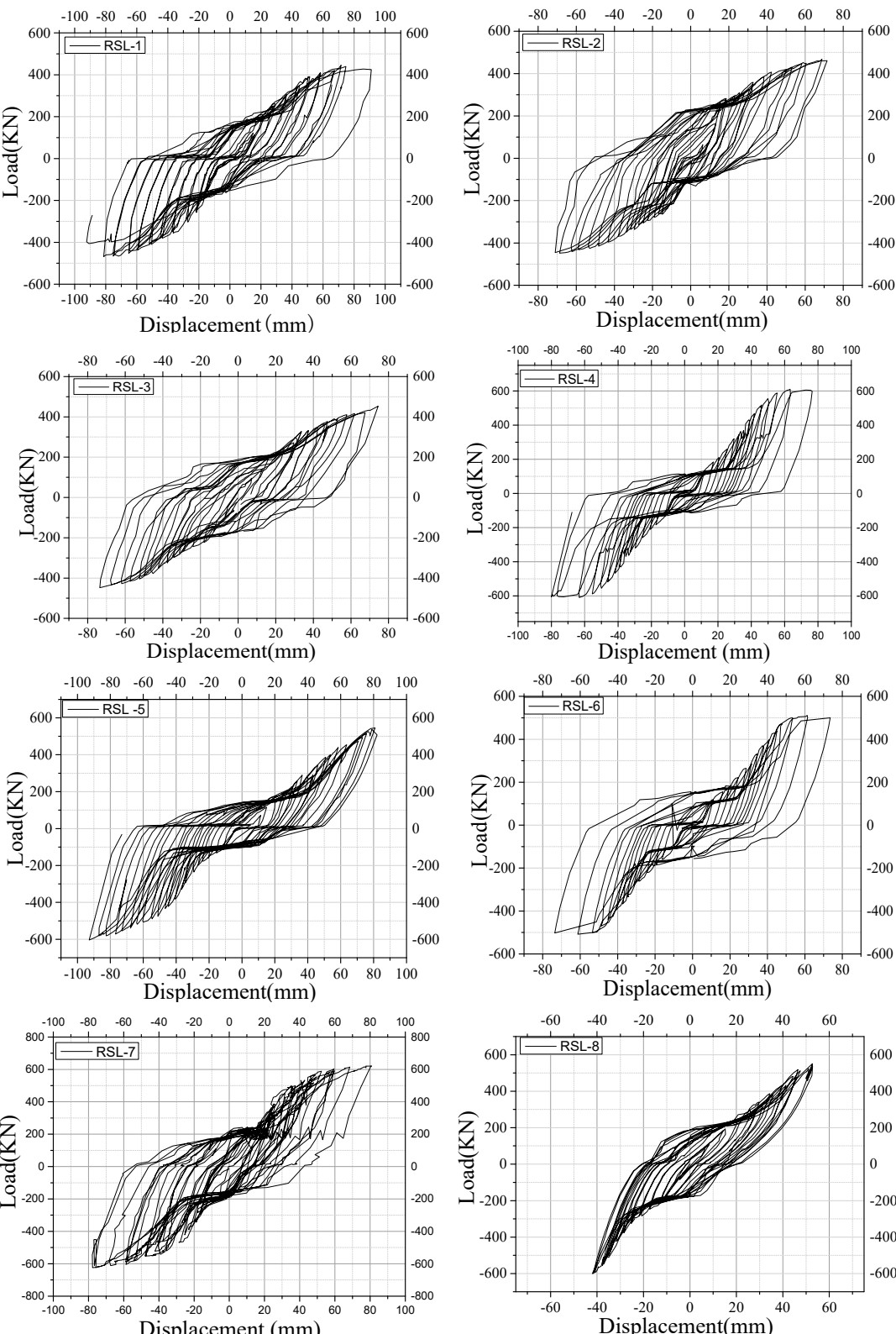

**Figure 11.** Hysteretic response of specimens.

As can be concluded from the above figures:

(1) The hysteretic curve shape roughly displays spindle shape, and it can be observed from the figure the hysteretic curve has three sections, including elastic section, yield section, and reinforcement

section. A certain displacement of rigid body has occurred during the test due to the loading equipment, resulting in a certain pinching of hysteresis curve.

(2) The mechanical properties of energy dissipation links are consistent under the action of tensile and compressive loads. The hysteretic curves are symmetrical in tension and pressure, and the hysteretic curves take the shape of central symmetry.

(3) When the load is initially applied, the curve is a straight line under the action of tension and pressure. When the displacement and load increase, the link starts to enter the yield from the elastic stage. The stiffness of the specimen decreases and the link produces plastic deformation. The area of hysteresis loop gradually increases with the increase of load, and the energy dissipation capacity of the link is gradually developed. The energy dissipation is stable at this stage. When the displacement reaches the limit value, the bearing capacity of the energy-dissipated link begins to decrease and the area of hysteresis loop reaches its maximum.

### 3.4. Energy Dissipation Behavior

$$E = \frac{S_1}{S_{\triangle AOB} + S_{\triangle COD}}$$

$S_1$ is the area of the maximum hysteresis loop.

The energy dissipation coefficient of each test specimen can be calculated by the hysteresis curve. Figure 12 is the schematic diagram of energy dissipation coefficient E. Table 6 shows the energy dissipation coefficient of each test specimen.

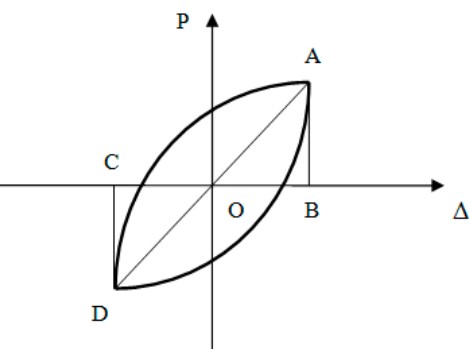

**Figure 12.** Calculation diagram of energy dissipation coefficient.

**Table 6.** Energy dissipation coefficient.

| Number | RSL-1 | RSL-2 | RSL-3 | RSL-4 | RSL-5 | RSL-6 | RSL-7 | RSL-8 |
|--------|-------|-------|-------|-------|-------|-------|-------|-------|
| E | 1.96 | 1.63 | 1.59 | 1.70 | 1.72 | 1.65 | 1.66 | 1.62 |

(1) The energy dissipation coefficient of each test specimen is relatively large, with the minimum energy dissipation coefficient being 1.59 of rsl-6 and the maximum being 1.96 of rsl-1, which indicated that the abdominal board-type replaceable energy dissipation link is able to dissipate the energy well in earthquake and it has good seismic performance.

(2) By comparing rsl-4, rsl-7 with rsl-8, it can be seen that when the link length, stiffening rib spacing, and yield type are the same, and the energy dissipation coefficient of the link with a large section size is relatively low.

(3) By comparing rsl-1, rsl-4 with rsl-6, the energy dissipation coefficient decreases when the length of the energy dissipation link increases, and it can be seen that the energy dissipation capacity of shear link is better than that of bending link.

(4) By comparing rsl-4 to rsl-5 in shear yield, when the stiffened ribs spacing between rsl-4 and rsl-5 decreases by 52.47%, the energy dissipation coefficient decreases by 0.74%. By comparison of rsl-5 and rsl-6 in the bending yield, rsl-5 decreases by 52.47% as compared with rsl-6 stiffening rib spacing, but the energy dissipation coefficient decreases by 2.45%. The energy dissipation coefficient of stiffening ribs decreases when the space between them is reduced.

(5) By comparing the energy dissipation coefficients of rsl-1, rsl-4, rsl-5, rsl-7, and rsl-8, the energy dissipation coefficients of the link decrease with the increase of $\zeta$, and the smaller the cross section of the link is, the higher the energy dissipation coefficient of the link.

### 3.5. Skeleton Curves

The skeleton curve of each specimen is obtained through the peak points of each hysteresis loop in the hysteretic curve, and Figure 13 shows the skeleton curve of each specimen.

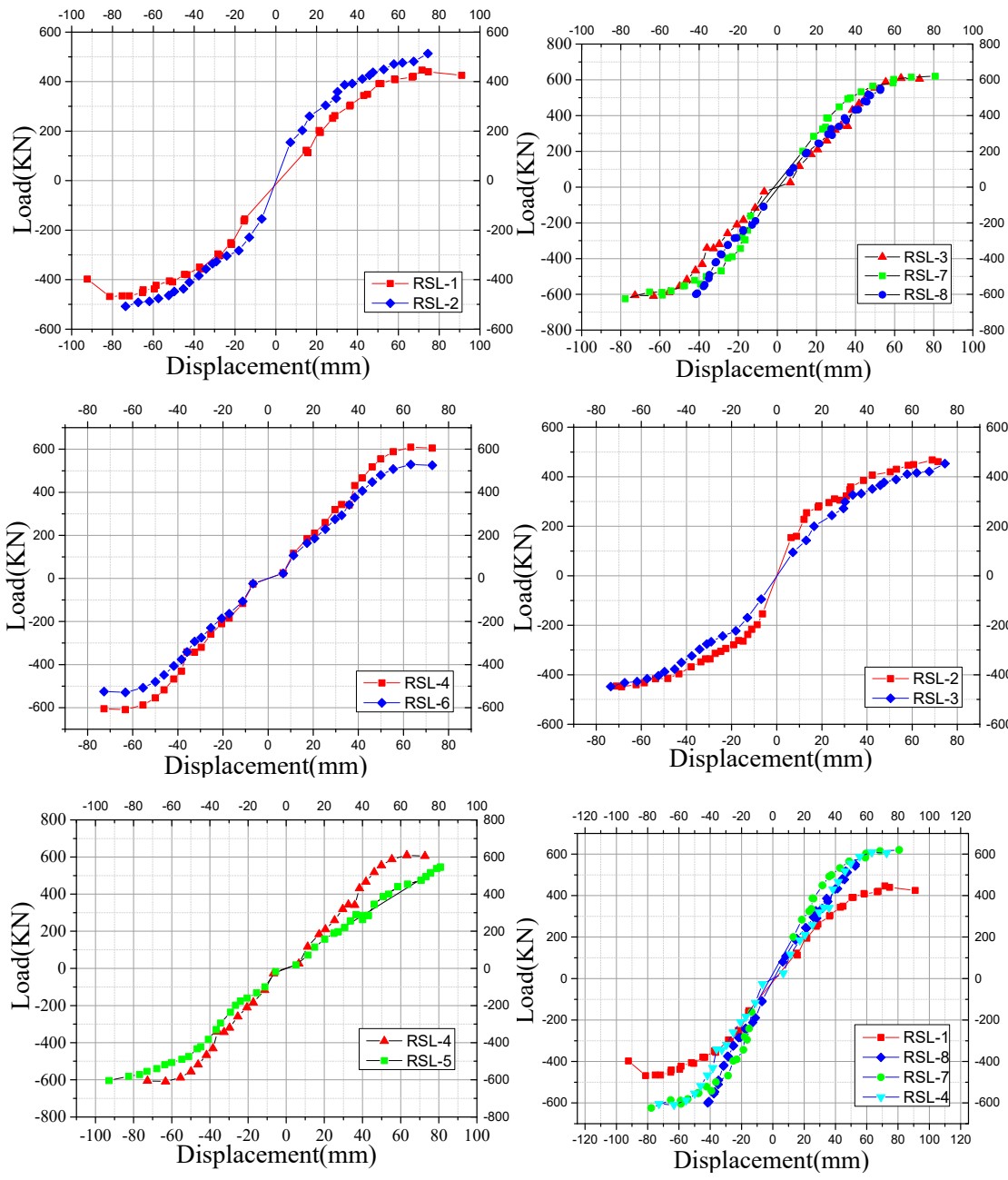

**Figure 13.** Skeleton curves of specimens.

The following conclusions can be obtained by the skeleton curve of the energy dissipation links.

(1)　As can be seen from the shape of the skeleton curve, the skeleton curve of all the specimens keeps roughly the same trend. It can be seen that the elastic section, yield section, and strengthening section are easy to observe. With the continuous increase of displacement, the stiffness of specimens shows a gradual downward trend.

(2)　By comparison of rsl-4, rsl-7, and rsl-8, it can be seen that, when the link length, stiffening rib spacing and yield type are the same, the skeleton curve of the link with relatively large section size has high bearing capacity and high stiffness. It can be seen that the skeleton curve of web connection type link is greatly affected by the size of the section, and the bearing capacity of the link with a larger size of the section is stronger.

(3)　By comparing rsl-1 with rsl-6, and rsl-4 with rsl-6, respectively, it can be seen that the skeleton curve has the same trend. The load peak of rsl-1 and rsl-6 differs by 15kN, and the skeleton curve of rsl-4 and rsl-6 are consistent. Therefore, when the cross-section size, number of stiffening ribs, and yield type of the replaceable link are the same, the link length has relatively little impact on the bearing capacity.

(4)　By comparing rsl-5 with rsl-6, and rsl-4 with rsl-5, respectively, it can be seen that the number of stiffening ribs of rsl-5 increases compared with rsl-6, but the skeleton curve has no change in trend and bearing capacity, and the curve is consistent. Therefore, it can be concluded that the stiffening spacing of the link has little influence on the skeleton curve when the cross-section size, length, and yield type of the link are the same.

(5)　As the size and length of the section of the link have impact on the changes of Mp and Vp of the link, it is necessary to compare the difference $\zeta$ between 1.16 and 1.57 as a variable on the basis of the same number of stiffening ribs of the link. By comparing the skeleton curves of rsl-1, rsl-8, rsl-7, with rsl-4, the larger the section size is, the higher the bearing capacity of the skeleton curve. Therefore, the difference of $\zeta$ does not determine the bearing capacity and deformation of the skeleton curve.

It can be seen from above that the section size is the parameter with the greatest influence on the skeleton curve.

*3.6. Stiffness Degradation*

Stiffness degradation is an important facet for the analysis of mechanical properties of replaceable links. The calculation formula is as follows:

$$K_{i} = \frac{\left|P_{i}^{+}\right| + \left|P_{i}^{-}\right|}{\left|\Delta_{i}^{+}\right| + \left|\Delta_{i}^{-}\right|} \tag{6}$$

where, $K_{i}$ is the stiffness at $i$ loading cycle; $P_{i}^{+}$ and $P_{i}^{-}$ are, respectively, the maximum load at the positive and negative directions at $i$ loading cycle; and, $\Delta_{i}^{+}$ and $\Delta_{i}^{-}$ are, respectively, the maximum displacement at both the positive and negative directions at $i$ loading cycle. The stiffness degradation amplitude of all specimens is shown in Table 7.

**Table 7.** Stiffness degradation amplitude.

| Number | RSL-1 | RSL-2 | RSL-3 | RSL-4 | RSL-5 | RSL-6 | RSL-7 | RSL-8 |
|---|---|---|---|---|---|---|---|---|
| SDM | 65.46% | 60.06% | 62.04% | 61.83% | 64.31% | 58.72% | 60.01% | 57.12% |

The following conclusions about stiffness degradation of link can be obtained from Table 7:

(1) Energy dissipation link cross section comparison:

By comparing RSL-4, RSL-7, with RSL-8, the stiffness degradation amplitude of link with larger section size is relatively lower, which indicates that the increase of section size results in a reduction of stiffness degradation amplitude and plastic deformation capacity of link.

(2) Energy dissipation link length comparison:

By comparing RSL-1, RSL-3, RSL-4, with RSL-6, with the increase of link length and the reduction of stiffness degradation, it can be seen that the plastic deformation capacity of the shear links is better than that of bending links.

(3) Energy dissipation link stiffener spacing comparison:

When the cross-section size, length, and yield type of the link are the same, the stiffening spacing of the link has little influence on the stiffness degradation amplitude. The stiffness degradation amplitude increases with increasing stiffening distance of the link.

### 3.7. Plastic Rotation Angle (Link Rotation Capacity)

The maximum bearing capacity, maximum displacement, and maximum rotation angle of the specimen can be obtained through the hysteretic curve, as shown in Table 8.

**Table 8.** Maximum response table.

| Number | RSL-1 | RSL-2 | RSL-3 | RSL-4 | RSL-5 | RSL-6 | RSL-7 | RSL-8 |
|---|---|---|---|---|---|---|---|---|
| Displacement | 81.35 | 71.5 | 74.5 | 63.25 | 82.6 | 73.55 | 77.72 | 52.75 |
| Angle | 0.36 | 0.22 | 0.23 | 0.19 | 0.25 | 0.18 | 0.24 | 0.16 |
| Bearing capacity | 468.33 | 461.00 | 453.33 | 529.33 | 545.33 | 511.00 | 449.33 | 550.67 |

The following conclusions can be drawn:

(1) By comparing RSL-4, RSL-7, with RSL-8, the bearing capacity and stiffness of the link with relatively larger section size are stronger when the length of the link, the spacing of the stiffeners and the yield type are the same.

(2) By comparing RSL-1, RSL-3, RSL-4, with RSL-7, the link length and the stiffening distance has little effect on the bearing capacity, when the section size and yield type of the replaceable energy dissipation links are the same.

(3) The minimum rotation angle of all specimens in the test was 0.16 rad, which indicated that the replaceable link has good rotational capacity.

(4) By comparing RSL-2, RSL-3, RSL-4, with RSL-5, on the condition that the section size, length, and yield type of energy dissipating links are the same, the stiffening spacing of energy-dissipating links has little influence on the bearing capacity.

## 4. Finite Element Analysis

### 4.1. Verification of Finite Element Models

In the finite element simulation, the deformation of the replaceable energy-dissipating link is mainly concentrated on the first row of bolt holes that were connected to the frame beam.

The flange at this position is obviously deformed and severely bent. Under the action of cyclic load, the bolt hole at the web position is seriously deformed, and the original circular hole becomes irregular (see Figure 14c). At the same time, a certain out-of-plane deformation occurs to the energy-consuming connection, and the deformation amount is small (see Figure 14d).

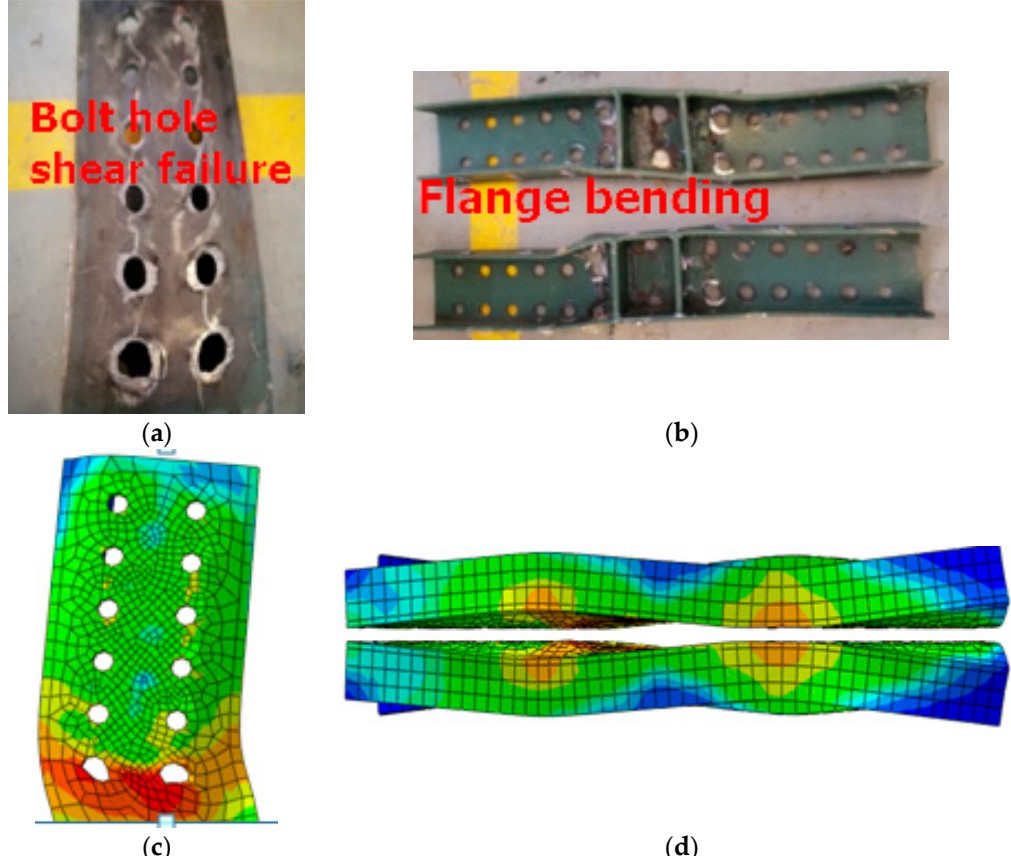

**Figure 14.** Test and finite element deformation comparison diagram. (**a**) bolt hole shear failure of test; (**b**) flange bending of test; (**c**) bolt hole shear failure of FEA; (**d**) flange bending of FEA.

In the test, the deformation of the replaceable energy-dissipating connection is mainly concentrated on the web and flange near the first row of bolt holes connected to the frame beam.

The bolt hole on the web is shear deformed. The deformation of the first row of bolt holes is more severe than that of the other bolt holes (see Figure 14a) and the flange is bent (see Figure 14b). Moreover, there is a certain out-of-plane deformation. This deformation is small due to the constraint of bolts.

The stress distribution law of finite element simulation is basically the same as shown in the test results, and the deformation position is consistent. The uniformity of test and finite element simulation results is fully proved.

*4.2. Finite Element Type and Mesh Size*

The ABAQUS multi-purpose finite element modelling code was used for the numerical modelling (Figure 15) of the test specimens (Table 1) of theWeb-connected links.

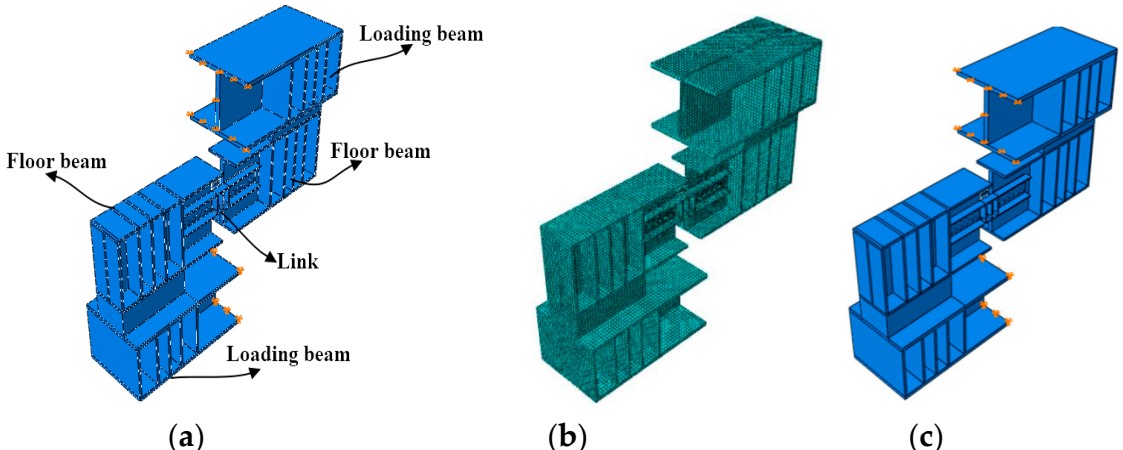

**Figure 15.** Finite element model (**a**); Mesh division diagram (**b**); and, Finite element model constraint diagram (**c**).

The loading device is also built into the finite element model to ensure that the actual stress of the link is the same as that of the test. The eight-node hexahedron quadratic reduction integral element (C3D8R) was used for the model. The loading equipment was composed of Q345B steel, which would not deform during loading due to its high rigidity. Subsequently, the loading beam and the frame beam were connected as a whole through tie. The link and loading beam, bolt, and frame beam were set as surface-to-surface contact, channel steel, and stiffener were connected through tie.

### 4.3. Loading and Boundary Conditions

The upper loading beam is restrained by the displacement in X and Z directions, and the Y-direction displacement is released for displacement loading, so as to simulate the left hinge support of the loading frame in the test. The beam that was loaded on the lower side constrains the displacement in X, Y, and Z directions, and the hinge bearing on the right side is simulated in the test. The meshing of link and high-strength bolts is relatively fine, so as to achieve the purpose of comparison with the test situation. The mesh spacing between the loading beam and the frame beam in the loading device is large. The stress cloud diagram of finite element is obtained when the cyclic load of equal amplitude is applied to the finite element model.

### 4.4. The Result Analysis of Test and Finite Element

As shown in Figure 16, the hysteresis curve of the finite element analysis is more full with a "shuttle" shape and the plastic deformation ability is better than the experimental results, but the overall fit is good. As shown in the figure above, when the beam section begins to yield under cyclic load, the hysteretic curve bends, the area of hysteretic loop increases, and the stiffness of the structure slightly decreases. With the increasing load, the residual deformation of energy-dissipating link continuously increases, the hysteresis loop tends to be full and stable, and the maximum hysteresis loop area is large without pinched, which indicates that the replaceable energy-dissipating link has strong plastic deformation capacity and good energy-dissipating performance.

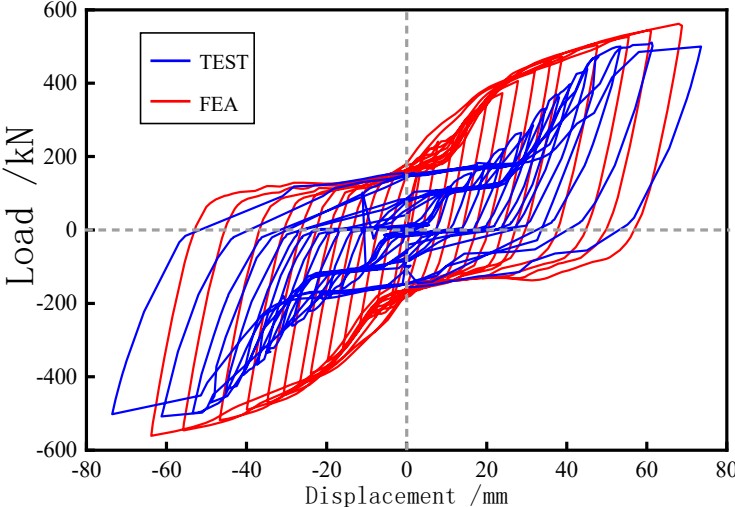

**Figure 16.** Comparison of hysteretic curves between test and finite element analysis of RSL-7.

It can be seen from Table 9 that the test results and finite element simulation results have a good degree of fitting, which can achieve mutual verification. Although there surely exists a difference in the bearing capacity, and the maximum difference even reaches 30.44% (this is because the specimen RSL-8 has some defects due to processing). On the other hand, the minimum difference is 1.00%, and when removed the accidental factors in the test, the difference between the test and the finite element bearing capacity is within 10%, which can meet the research requirements.

**Table 9.** Comparison table of test and finite element.

| Number | Test/kN | Finite Element /kN | Difference Value |
|--------|---------|--------------------|------------------|
| RSL-1 | 468.33 | 463.66 | 1.00% |
| RSL-2 | 467.66 | 461.77 | 1.26% |
| RSL-3 | 453.33 | 448.81 | 1.00% |
| RSL-4 | 609.33 | 664.40 | 8.29% |
| RSL-5 | 604.00 | 670.48 | 9.91% |
| RSL-6 | 529.33 | 561.59 | 5.75% |
| RSL-7 | 624.33 | 682.43 | 8.81% |
| RSL-8 | 550.67 | 791.71 | 30.44% |

*4.5. Stress Distribution*

Figure 17, respectively, show the overall stress cloud diagram of the specimen, the bolt stress cloud diagram, the Mises stress cloud diagram, and the shear stress cloud diagram.

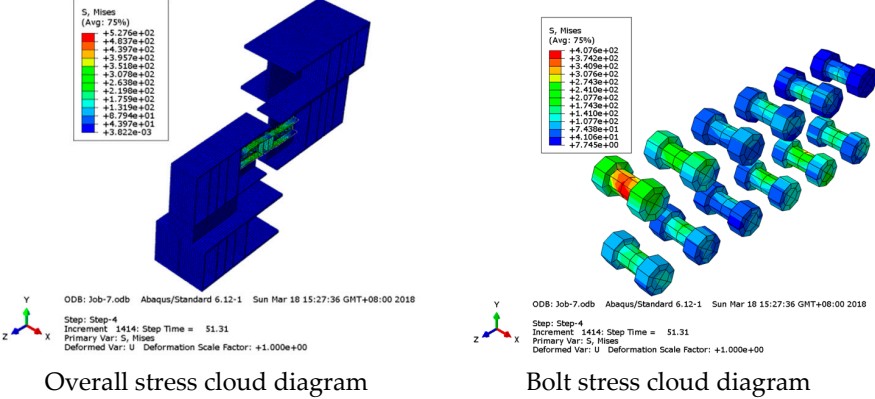

Overall stress cloud diagram          Bolt stress cloud diagram

**Figure 17.** *Cont.*

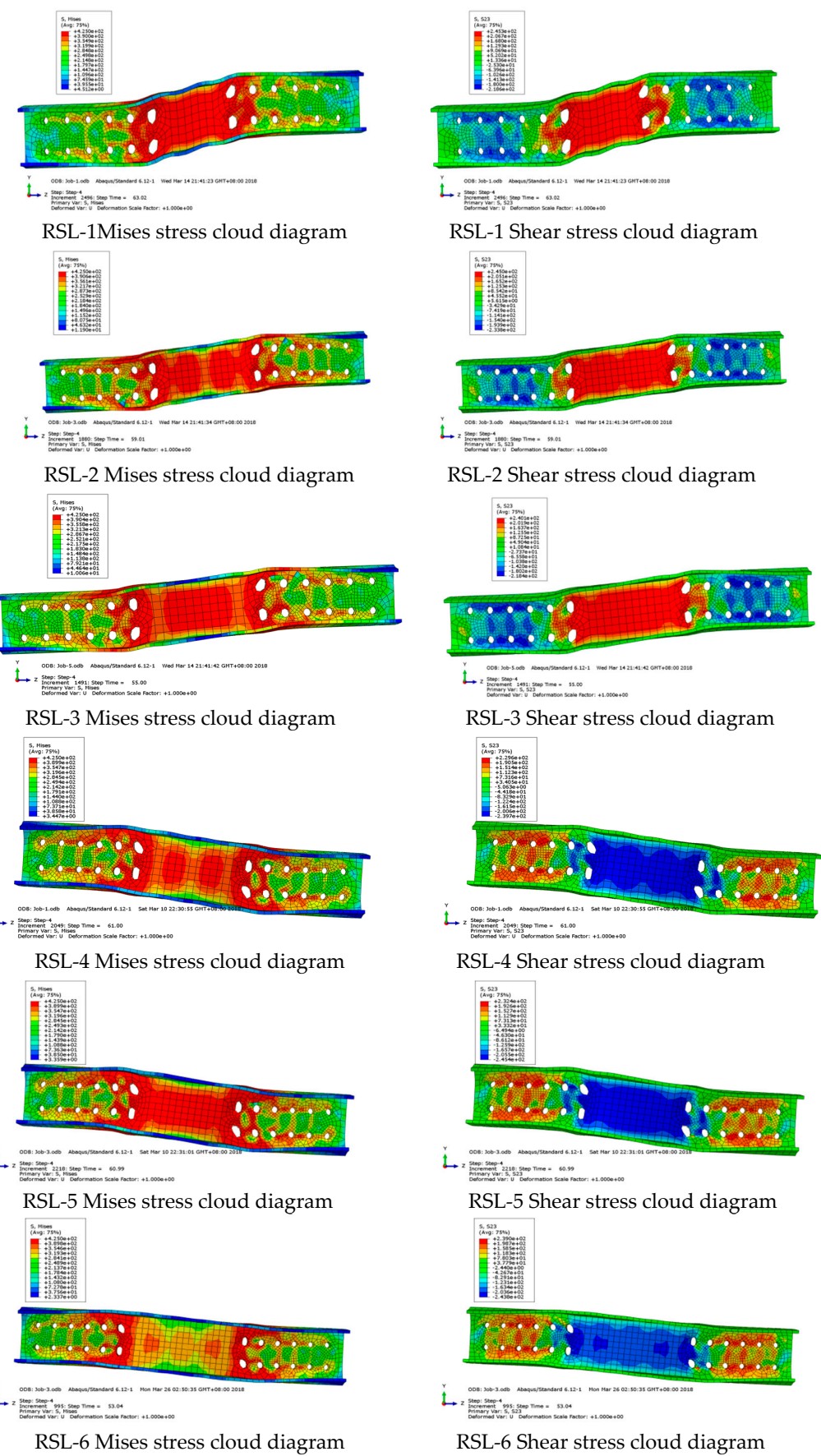

Figure 17. *Cont.*

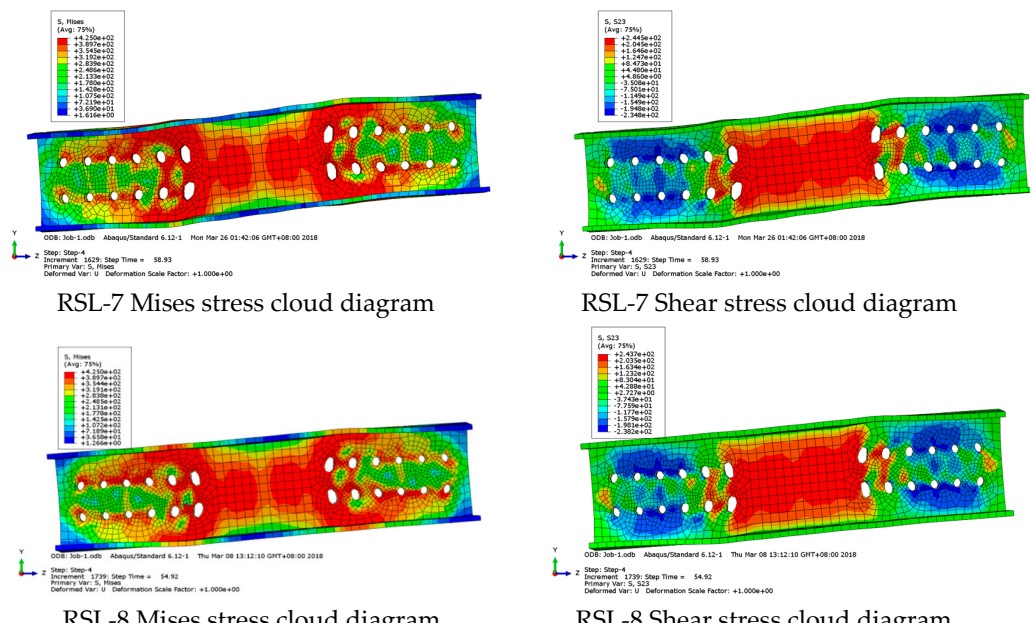

RSL-7 Mises stress cloud diagram  RSL-7 Shear stress cloud diagram

RSL-8 Mises stress cloud diagram  RSL-8 Shear stress cloud diagram

**Figure 17.** Stress cloud diagram of each specimen under cyclic load.

From the overall stress cloud diagram of the specimens, except for bolts, the maximum stress is concentrated on the energy-dissipated link, and the stress on other components, such as loading beam, frame beam, and high-strength bolt is very small. All of the above-mentioned parts are in the elastic range, and the deformation is mainly concentrated on the replaceable energy-dissipated link.

## 5. Conclusions

The performance of the replaceable energy dissipation link with web-connected is experimentally evaluated in this paper. This paper is aimed at a comprehensive and detailed analysis of link performance from the aspects of stress distribution hysteretic curve, skeleton curve, energy dissipation coefficient, stiffness degradation, and angle of rotation. The shear yield area of the link web is developed along the diagonal direction of the web under the action of tension and pressure. When the diagonal direction shear yield area is transfixion, it expands from the diagonal as the baseline to both sides until the whole web shear yield. When the shear stress on the web reaches the shear yield stress, the flange is still in the elastic stage. The tension and pressure on the hysteretic curve are basically symmetric, and the elastic section, yield section and strengthening section can be clearly observed. The cross section size of the link has a great influence on the skeleton curve. However, the stiffening distance and the length of the link have little effect on the skeleton curve. The energy dissipation coefficient decreases as $\zeta$ increases, the cross section of the link and $\zeta$ are relatively small, and the energy dissipation coefficient of the connection is relatively high. The energy dissipation coefficient of each specimen is relatively large, which stays between 1.59 and 1.96. The energy dissipation capacity of shear connection is better than that of curved link. The stiffening spacing has little effect on the stiffness degradation amplitude. The replaceable links have good rotational capacity, and the rotation angle of each specimen is 0.16 rad at the minimum. In addition, the failure mode of the finite element analysis and the test results are similar to each other, which can achieve mutual verification. The results show that the web-connected link has good energy dissipation performance.

In future work, the author will continue to complete some frame system tests and a lot of finite element analysis [20,50–53]. A comprehensive parametric computational study is conducted to investigate the main parameters and rules that affect the mechanical properties of links after validating the accuracy of the finite element (FE) modelling approach against previous experiments [54–58]. Further theoretical analysis need to do to establish the resilience model of the link. Additionally, in

further studies, the optimized design methodology based on the yielding criterion is proposed. From this research, we see the urgent need for crack monitoring and bolt loosening monitoring after a seismic event. However, the strain gauge sensors that were used cannot monitor these types of damages in real time. With the recent development of smart aggregates in structure damage monitoring [59–64] and the application of piezoceramic transducer transducer-based in health monitoring of steel structures [65–68], the authors will explore health monitoring and damage detection of eccentric braced steel frames while using the easy-to-operate smart aggregate and easy-to-install piezoceramic patch transducers in the future work.

**Author Contributions:** Z.Y. designed the experiments. Z.H. performed the experiments. Z.Y., Z.H., D.F., H.Z. analyzed the data and wrote the paper.

**Funding:** This research is supported in part by National NSFC (Natural Science Foundation of China) (51568040, 51368037). Basic Research Foundation of Colleges and Universities of Gansu Province.

**Conflicts of Interest:** The authors declare no conflict of interest.

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
