# Peer review of "Experimental Study on Energy Dissipation Performance and Failure Mode of Web-Connected Replaceable Energy Dissipation Link"

_applsci, doi:10.3390/app9153200_

Round 1

Reviewer 1 Report

The authors investigated energy dissipation performance of web-connected replaceable energy dissipation links.     

·         The English syntax and grammar needs to be extensively improved. It is suggested to use English center service for improving the linguistic aspects of the article.  Also, many words are used that are not suitable context wise. These are a few examples:

o   it is difficult to repair or replace the link of energy dissipation

o   Meanwhile it will increasing

o   In order to solve the above-mentioned shortcomings in the design of the eccentrically braced frame structure

o   The results indicate clearly that the links of web-connected shows good ductility and stable energy dissipation ability.

o   specimens of 23 full-scale eccentric support frames, The connection

o   The hysteretic curve is roughly shaped like a fusiform,

o   abdominal board-type replaceable energy dissipation

o   future work, the author will do a lot of finite element parameter analysis to get some quantitative laws of the replaceable energy dissipation link with web-connected

·         Many vague statements that are due to poor English grammatical errors rather than technical issues. Vague/irrelevant remarks in introduction. These are a few examples:

o   According to the relative relation between the link's plastic moment and shear ability, the link length can be divided into short links where the shear yielding of the web is found to be the dominant factor and long links where the flexural yielding controls the link performance

o   What is length ratio defined in introduction?

·         In key words, the “eccentrically braced frame” should be added.

·         The statement “Specimens L1, L4, and L5, L7, L8 failed by local buckling of the flange after shear yield of the web, which was likely caused by the energy dissipation of the link web yield coupled with the flexure yield of the flange”  is  not accurate.

·         Many terms such as curved link, ζ, etc. should be clearly defined.

·         What conclusion is made based on comparison of the specimens with each other? This conclusion should be added before section 3.2

·         The purpose of the article is unclear. The idea of having shear links/fuses haven been worked on in many studies; therefore, the authors should clearly state their contribution and the purpose of their work in a paragraph or two.

·         The initial design and selection of the section seems to be random. Why these sections are used? based on which concept? And for what purpose?

·         The authors are encouraged to consider recent developments on Eccentrically braced system. Some new works are listed below:

o   Farzampour, A., Mansouri, I., & Dehghani, H. (2019). Incremental Dynamic Analysis for Estimating Seismic Performance of Multi-Story Buildings with Butterfly-Shaped Structural Dampers. Buildings, 9(4), 78.

o   Farzampour, A., & Eatherton, M. (2019). Parametric Computational Study on Butterfly-Shaped Hysteretic Dampers.

·         Define each term used. Examples are:

o   thickness ratio of the web,

o   the width ratio of the flange

o   Table 1  fourth column

·         What code is the loading protocol based on?

·         Table 3 and Table 4 do not show much of information. If these table are included, the conclusion based in these table should be elaborated.

·         The authors mentioned “The energy dissipation coefficient of stiffening ribs decreases when the space between them is reduced”, is that correct? Is there any other work that corroborate this statement?

·         More details on finite element modeling methodology are needed (how the load is applied, boundary condition, constitutive model, mesh size, mesh sensitivity, etc.)

·         Section 4.4 should be the first section in finite element section. ( verification before results)

·         For having a complete verification,  it is needed to obtain finite element hysteretic results ( at least for one model) and compare it with one of the figures shown in Figure 1. The authors have only compared the general mode of behavior which shows an incomplete verification.

·         For references: first the authors should follow the guidelines carefully ( Format and Font) and too many works of the same author should be avoided. Examples are :

o   Yin, Z.Z.; Feng D.Z.; Yang, W.W. Damage analyses of replaceable links in eccentrically braced frame (EBF) subject to cyclic loading. Appl. Sci. 2019, 9, 332.

o   Yin, Z.Z.; Zhang, H.; Yang, W.W. Study on seismic performance analysis of steel plate shear wall with partially encased composite (PEC) columns. Appl. Sci. 2019, 9, 907.

o   Yin, Z.Z.; Bu F.C. Overall stability analysis of improved buckling restrained braces. The Open Civil Eng. J.

o   Yin, Z.Z.; Ren Y.G.; Chen, W.; Liang, Y.X.The seismic performance analysis of replaceable independent links.J. Eng. Mech. 2016, 33, 207-213.

Or

o   Mansour, N. Eccentrically braced frames with replaceable shear links. Ph.D. Thesis, Department of Civil Engineering, University of Toronto. 2010, Toronto, Ontario, Canada.

o   Mansour, N.; Christopoulos, C.; Tremblay, R. Experimental validation of replaceable shear links for

o   Mansour, N. Eccentrically braced frames with replaceable shear links. Ph.D. Thesis, Department of Civil Engineering, University of Toronto, Toronto, Ontario, Canada, 2010.

Author Response

Thank you for your letter and for the reviewers’ comments concerning our manuscript entitled “Experimental Study on Energy Dissipation Performance and Failure Mode of Web-Connected Replaceable Energy Dissipation Link” (ID: applsci-532652). Those comments are all valuable and very helpful for revising and improving our paper, as well as the important guiding significance to our researches. We have studied comments carefully and have made correction which we hope meet with approval. Please see the attachment.

Reviewer 2 Report

The submitted manuscript illustrates an experimental and numerical study involving steel eccentric braces where the dissipative tract has been re-designed to allow its replacement when damaged after seismic events. The subject can be considered as interesting for the structural engineering community involved into seismic design of steel structures. However, there are a number of major issues that must be addressed before the submitted manuscript could be suggested for publication.

1) The first major issue is the very poor quality of the presentation, unacceptable for any kind of publication, let alone a publication as a journal article. Reading is a difficult and fatiguing experience, and many sentences cannot be fully understood. It should be absolutely enforced the help of an experienced support to improve English to a decent level.

2) The Authors do not give a sound proof that link replacement is actually possible after plastic damage has occurred and residual interstorey drifts have displaced horizontally the steel frame. Please clarify this point and provide substantial information, experimental results, and an actual proof that replacement is possible.

3) The state of the art on the subject appears limited. While attention to replaceable steel links in seismic constructions is limited, there are many researches that in the past two decades have studied replaceable steel links in steel-concrete hybrid structures. The Authors are recommended to have a look at Scopus or other scientific search engines and improve their state of the art accordingly.  

Author Response

(The authors gave the same response as above.)

Reviewer 3 Report

Language of the paper is poor, makes substance unclear and confusing. The experimental results are as expected from current understanding, so impact is limited. The numerical modelling looks realistic. The work appears to be ongoing, further finding may be expected. There are some problems in formatting (e.g. table3 in page 10-11, mismatch fonts in Reference 41).

Author Response

(The authors gave the same response as above.)

Round 2

Reviewer 1 Report

The authors have answered to majority of the comments. However, the article needs to be revised by English Service center due to many syntax and grammatical issues. The technical quality of the work is desirable, but the authors should extensively improve the English grammatical and syntax quality of the work.

·         The authors did a verification study in Fig. 17. The two curves show significant difference.  This verification is not acceptable, since the majority of the cycles are off by more than 30%. it does not hit the right drift value. The authors are recommended to improve the verification method they use in their work and check if the computational models are valid accordingly.

·         Verification should be placed before computational results. Also, the table 8 which is based on the Figure 17 is not clear. Since the differences between the two curves in Figure 17 are significantly more than 1%, which is shown in Table 8

·         The authors are encouraged to compare their work to new EBF systems, or at least show a plan for doing so. In this regard the authors are recommended to consider these following references in their work to show possible comparison with different new EBF designs, and possible alternatives.

o   Farzampour, A., Mansouri, I., & Dehghani, H. (2019). Incremental Dynamic Analysis for Estimating Seismic Performance of Multi-Story Buildings with Butterfly-Shaped Structural Dampers. Buildings, 9(4), 78.

o   Farzampour, A., & Eatherton, M. (2019). Parametric Computational Study on Butterfly-Shaped Hysteretic Dampers.

o   Farzampour, A. (2019). Evaluating Shear links for Use in Seismic Structural Fuses (Doctoral dissertation, Virginia Tech).

.

·         The quality of figures should be improved. Legends are unreadable, the description under some figures are unclear (e.g. Figure 16)

·         Figure Explanations should be added before the figures.

Author Response

Response to Reviewer 1 Comments

Dear Reviewer:
Thanks very much for your hard work. Those comments are all valuable and very helpful for improving our paper, as well as the important guiding significance to our researches. We have studied comments carefully and have made correction which we hope meet with approval. Revisions are clearly highlighted in the paper. Thank you very much for all your help and looking forward to hearing from you soon.

Best regards

Sincerely yours

The main corrections in the paper and the responses to the reviewer’s comments are as follows:

Comment 1: The authors have answered to majority of the comments. However, the article needs to be revised by English Service center due to many syntax and grammatical issues. The technical quality of the work is desirable, but the authors should extensively improve the English grammatical and syntax quality of the work.

Response 1: Thank you very much for your requirements in language improvement. We are very sorry for inexact and misleading expressions in English. We have asked help from several English teachers to improve syntax and grammatical issues. We have carefully revised the manuscript, especially rewritten the parts of abstract, introduction and conclusion with the corresponding revision part being highlighted the in red. We hope the revised article can be more readable and accurate.

Comment 2: The authors did a verification study in Fig. 17. The two curves show significant difference.  This verification is not acceptable, since the majority of the cycles are off by more than 30%. It does not hit the right drift value. The authors are recommended to improve the verification method they use in their work and check if the computational models are valid accordingly.

Response 2: Some of the test results were compared with finite element simulation results, such as deformation of specimen, stress distribution, energy dissipation capacity and so on. The comparison results show that the test results and finite element simulation results fit well and can be verified by each other, though there will be some slightly disparities. Fig. 17 (according to the revised version, it should be fig.16) shows the comparison of energy dissipation capacity of specimen RSL7. The reasons why the two curves are not completely consistent are as follows: 1) the specimen material properties in the test and numerical simulation are not completely consistent. One is measured in practical condition and the other is theoretical; 2) there are some deviations between theory and practice, such as deviations of geometric dimensions and loading conditions. The disparity of most specimens is relatively small, and only the specimen RSL-8 has a large difference in bearing capacity. This is because the specimen RSL-8 has some defects due to processing. In order to truly reflect the test phenomenon, we also site this result in the paper and analyse the causes. Therefore, in the follow-up study, we need to further advance the test method and the computational models to minimize the disparities as much as possible.

Comment 3: Verification should be placed before computational results. Also, the table 8 which is based on the Figure 17 is not clear. Since the differences between the two curves in Figure 17 are significantly more than 1%, which is shown in Table 8.

Response 3: The verification section has been placed before the calculation results, according to the review comments.The Table 8 and the Figure 17 displays different contents. The Table 8 shows the comparison of the maximum load capacity of the specimens in the test and numerical simulation. The difference between the test and the finite element bearing capacity is within 10%, which can meet the research requirements. Figure 17 (according to the revised version, it should be fig.16) shows the comparison of energy dissipation capacity of specimen RSL7. The comparison results of other specimens are similar and are not listed in this paper. The results show that the hysteretic properties of both the test and finite element are stable and the shape of hysteretic curve is basically the same. The hysteretic curve in the test is not as full as the finite element for the reasons described in response 2.

Comment 4: The authors are encouraged to compare their work to new EBF systems, or at least show a plan for doing so. In this regard the authors are recommended to consider these following references in their work to show possible comparison with different new EBF designs, and possible alternatives.

o   Farzampour, A., Mansouri, I., & Dehghani, H. (2019). Incremental Dynamic Analysis for Estimating Seismic Performance of Multi-Story Buildings with Butterfly-Shaped Structural Dampers. Buildings, 9(4), 78.

o   Farzampour, A., & Eatherton, M. (2019). Parametric Computational Study on Butterfly-Shaped Hysteretic Dampers.

o   Farzampour, A. (2019). Evaluating Shear links for Use in Seismic Structural Fuses (Doctoral dissertation, Virginia Tech).

Response 4: We strongly agree with your suggestion, and have added this literature to our paper. Considering your suggestion, we have studied related literature carefully and compared our study with different new EBF designs by other researchers, thus some conclusions could be drawn:

Farzampour, A. and Mansouri, I. proposed a butterfly-shaped link that is to align better bending strength along the length of the link with the demand moment diagram. The butterfly-shaped link and the link we studied in the paper are all used to resist seismic force with shear loading resistance capabilities. The new systems are designed to protect the structural integrity and concentrate the inelasticity on a specific area, while the remaining parts are kept undamaged and intact. The damage and excessive inelastic deformations are concentrated on the link to avoid any issues for the rest of the surrounding elements [66-69]. In these literatures, the effect of shear and flexural stresses on the behaviour of butterfly-shaped links are computationally investigated. The mathematical models based on von-Mises yielding criteria are initially developed and the optimized design methodology is proposed based on the yielding criterion. In addition, a parametric computational study is conducted to investigate the shear yielding, flexural yielding, and lateral torsional buckling limit states for butterfly-shaped links. Some related literatures are also included in the references section in red in the paper.

Through comparison, we also put forward some ideas of the next step:

Firstly, we will continue to do some frame system tests and a lot of finite element analysis.

After validating the accuracy of the finite element (FE) modelling approach against previous experiments, a comprehensive parametric computational study is to be conducted to investigate the main parameters and rules that affect the mechanical properties of structures.

Secondly, we need to do further theoretical analysis to establish the resilience model of the link.

Finally, the optimized design methodology based on the yielding criterion is proposed.

66.     Farzampour, A.; Mansouri, I.; Dehghani, H. Incremental dynamic analysis for estimating seismic performance of multi-story buildings with butterfly-shaped structural dampers. Buildings, 2019, 9(4), 78.

67.     Farzampour, A., & Eatherton, M. Parametric Computational study on butterfly-shaped hysteretic dampers. Front. Struct. Civ. Eng. 2019, https://doi.org/10.1007/s11709-019-0550-6.

68.     Farzampour, A. Evaluating Shear links for Use in Seismic Structural Fuses. (Doctoral dissertation, Virginia Tech). 2019.

69.     Farzampour, A; Eatherton, M. Yielding and lateral torsional buckling limit states for butterfly-shaped shear links. Struct. Eng. 2019, 180, 442-451.

Comment 5: The quality of figures should be improved. Legends are unreadable, the description under some figures are unclear (e.g. Figure 16)

Response 5: We are very sorry that we did not treat the pictures carefully enough, making the legend unreadable. Considering your suggestion, we have rewritten this part and rearranged the figures and tables, adding it to the paper.

Comment 6: Figure Explanations should be added before the figures.

Response 6: According to the review comment, the figure explanations have been placed before the figures.

Reviewer 2 Report

The Authors failed to address the comments of this Reviewer: 1) the poor quality of the presentation was not improved; 2) there is not sound proof that link replacement is actually possible after plastic damage has occurred and residual interstorey drifts have displaced horizontally the steel frame; 3) no adequate improvements in the state of the art review were made. Given that the Authors are not able to address these comments, my recommendation is to reject the submitted manuscript.

Author Response

Response to Reviewer 2 Comments

Dear reviewer,

Thanks very much for your comments concerning our manuscript. Those comments are all valuable and very helpful for revising and improving our paper, as well as the important guiding significance to our researches. I apologize for my imperfect response to the reviewer’s comments (Round 1). We have once studied comments (Round 2) carefully and have made several corrections which we hope meet with approval. New revisions are clearly highlighted in the paper. Thank you very much for all your help and looking forward to hearing from you soon.

Best regards

Sincerely yours

The main corrections in the paper and the responds to the reviewer’s comments are as follows:

Comment 1: The poor quality of the presentation was not improved.

Response 1: Thank you very much for your requirements in language improvement.We are very sorry for inexact and misleading expressions in English. We have asked help from several English teachers to improve grammar and sentences. We have carefully revised the manuscript, especially rewritten the parts of abstract, introduction and conclusion with the corresponding revision part being highlighted the in red. We hope the revised article can be more readable and accurate.

Comment 2: There is not sound proof that link replacement is actually possible after plastic damage has occurred and residual interstorey drifts have displaced horizontally the steel frame.

Response 2: Thank you very much for your hard work. We greatly appreciate your valuable and very helpful comments. The replaceable link studied in this paper are used in EBF systems (Figure 1). Through a series of tests and a large number of finite element analysis (Figure 1), it can be known that under cyclic loading, the deformation of the specimens is concentrated on the replaceable links, and most of the stress reaches the ultimate strength of the materials. But the rest of the structure (frame column, frame beam, brace) remains elastic. The residual deformation of the frame is very small. According to the test data (Table 1), after plastic damage has occurred on the link, the residual deformation of the frame is very small. The maximum stress on frame beam, column and brace is very small, which is far less than the yield stress of the material. The above analysis shows that the frame beams columns and brace are still in the elastic stage after several energy dissipation links replacement. The link has a good replaceable properties. The high-strength bolt is removed from the original frame by an electric wrench when the link is broken. It is feasible to replace the link after adjusting to the original frame position with the jack, as shown in Figure 2.

Table 1.  The specimen plastic corner

specimen

RSL-1s

RSL-2s

RSL-3s

RSL-4s

plastic corner

0.230

0.170

0.140

0.080

specimen

RSL-5s

RSL-6s

RSL-7s

RSL-8s

plastic corner

0.100

0.150

0.060

0.050

Figure 1. The frame model

Figure 2. Replacement of energy-dissipating links

Comment 3: The state of the art on the subject appears limited. While attention to replaceable steel links in seismic constructions is limited, there are many researches that in the past two decades have studied replaceable steel links in steel-concrete hybrid structures. The Authors are recommended to have a look at Scopus or other scientific search engines and improve their state of the art accordingly.  

Response 3:  We greatly appreciate your valuable and very helpful comments for our experimental technique. Considering your suggestion, we have studied related literature carefully and compared our study with similar researches that have studied replaceable steel links in steel-concrete hybrid structures, and some conclusions could be drawn:

An hybrid coupled wall (HCW) system made of a single reinforced concrete (RC) wall coupled to two steel side columns by means of steel links is presented. The design objective is to reduce or possibly avoid the damage in the RC wall while concentrating the seismic damage on the replaceable steel links intended to be the only dissipative components of the presented HCW system [1]. Alper [2] studied a benchmark building frame with and without bolted dissipative beam splices. The performance of both innovative and conventional structures has been quantified in terms of energy dissipation, floor displacements and inter-story drifts, as a result of nonlinear transient dynamic analysis. In the Morelli’s paper[3] the development, calibration and experimental validation of two component-based models of dissipative steel links connecting a reinforced concrete wall to a steel gravity frame is presented. Replaceable steel coupling beams (RSCB) [4] have been proposed as an alternative to conventional reinforced concrete (RC) coupling beams for enhanced seismic resiliency of coupled wall systems. This paper presents a series of quasi-static tests conducted to examine the seismic behaviour of RSCBs with RC slabs and to identify reasonable slab configurations that can minimize the damage to RC slabs. The performance of both innovative and conventional structures has been quantified in terms of energy dissipation, floor displacements and inter-story drifts, as a result of nonlinear transient dynamic analysis.

In these new frame type and the link we studied in the paper, damage concentrates mainly on the bolted dissipative beam splices (links) acting as “structural fuses”, which can be easily and inexpensively replaced after strong seismic events.

According to the above analysis, some plans and adjustments have been made for the follow-up works:

Firstly, we will continue to do some frame system tests and a lot of finite element analysis. We need to further advance the test method and the computational models to minimize the disparities as much as possible. After validating the accuracy of the finite element (FE) modelling approach against previous experiments, a comprehensive parametric computational study is conducted to investigate the main parameters and rules that affect the mechanical properties of structures.

Secondly, we need to do further theoretical analysis to establish the resilience model of the link.

Finally, the optimized design methodology based on the yielding criterion is proposed.

Sincerely thank you again for your valuable comments.

1.          AlessandroZona   Hervé Degée   Graziano Leoni. Ductile design of innovative steel and concrete hybrid coupled walls. J. Constr. Steel. Res. 2016, 117, 204-213.

2.          Alper, K. Milot, M.; Influence of repairable bolted dissipative beam splices (structural fuses) on reducing the seismic vulnerability of steel-concrete composite frames. Soil. Dyn. Earthq. Eng. 2019, 117, 281-298.

3.          Morelli, F. Manfredi, M. Salvatore, W.;  An enhanced component based model for steel connection in a hybrid coupled shear wall structure: development, calibration and experimental validation. Computers & Structures. 2016, 176, 50-69.

4.          Alireza, F. Farhad, K. Pouya, P.; Experimental study of a replaceable steel coupling beam with an end-plate connection. J. Constr. Steel. Res. 2016, 122, 138-150.

Reviewer 3 Report

There are still some problems in language and formatting.

Author Response

Response to Reviewer 3 Comments

Dear Reviewer:
Thanks very much for your hard work. Those comments are all valuable and very helpful for improving our paper, as well as the important guiding significance to our researches. We have studied comments carefully and have made correction which we hope meet with approval. Revisions are clearly highlighted in the paper. Thank you very much for all your help and looking forward to hearing from you soon.

Best regards

Sincerely yours

Response to comment 1: There are still some problems in language and formatting.

Response 1: Thank you very much for your requirements in language improvement. We are very sorry for inexact and misleading expressions in English. We have asked help from several English teachers to improve grammar and sentences. We have carefully revised the manuscript, especially rewritten the parts of abstract, introduction and conclusion with the corresponding revision part being highlighted the in red. We hope the revised article can be more readable and accurate.

Round 3

Reviewer 2 Report

The Authors have to be congratulated for their effort in improving their manuscript. The quality of the presentation has reached a level adequate for a journal publication. Regarding the second and third issues pointed out by this Reviewer (proof of link replacement and state of the art review), the Authors have provided detailed and convincing response. However, this Reviewer cannot find any trace of such responses in the revised version of the manuscript. It is very important that the details provided in the Author Response file are incorporated in the manuscript, including Table 1 (the specimen plastic corner) and Figure 2 (Replacement of energy-dissipating links).  

Author Response

Dear reviewer Thank you very much for your suggestions for further improvement of our manuscript. Considering your suggestion, we made further revisions to our manuscript. The main corrections in the paper and the response to the reviewer’s comments are as follows: Comment 1: The Authors have to be congratulated for their effort in improving their manuscript. The quality of the presentation has reached a level adequate for a journal publication. Regarding the second and third issues pointed out by this Reviewer (proof of link replacement and state of the art review), the Authors have provided detailed and convincing response. However, this Reviewer cannot find any trace of such responses in the revised version of the manuscript. It is very important that the details provided in the Author Response file are incorporated in the manuscript, including Table 1 (the specimen plastic corner) and Figure 2 (Replacement of energy-dissipating links). Response 1: Thanks very much for your hard work. We strongly agree with your suggestion, and have added this part to our paper (line 227-244).New revisions are clearly highlighted in the paper. Through the tests and a large number of finite element analysis, it can be known that under cyclic loading, the deformation of the specimens is concentrated on the replaceable links, and most of the stress reaches the ultimate strength of the materials. But the rest of the structure (frame column, frame beam, brace) remains elastic. The residual deformation of the frame is very small. According to the test data (Table 3), after plastic damage has occurred on the link, the residual deformation of the frame is very small. The maximum stress on frame beam, column and brace is very small, which is far less than the yield stress of the material. The above analysis shows that the frame beams columns and brace are still in the elastic stage after several energy dissipation links replacement. The link has a good replaceable properties. The high-strength bolt is removed from the original frame by an electric wrench when the link is broken. It is feasible to replace the link after adjusting to the original frame position with the jack, as shown in Figure10. Table 3. The specimen plastic corner specimen RSL-1s RSL-2s RSL-3s RSL-4s plastic corner 0.230 0.170 0.140 0.080 specimen RSL-5s RSL-6s RSL-7s RSL-8s plastic corner 0.100 0.150 0.060 0.050 (a) (b) Figure 10. Replacement of energy-dissipating links Sincerely thank you very much for all your help and look forward to hearing from you soon. Best regards

Round 4

Reviewer 2 Report

The Authors have addressed most of the comments of the previous review round. However, it is not clear why the additional information provided in the Authors' response to report 2 have not been added in revised version of the manuscript. For reference, hereafter the copy-and-past from the response of the Authors that has not been implemented in the text of the revised manuscript:

An hybrid coupled wall (HCW) system made of a single reinforced concrete (RC) wall coupled to two steel side columns by means of steel links is presented. The design objective is to reduce or possibly avoid the damage in the RC wall while concentrating the seismic damage on the replaceable steel links intended to be the only dissipative components of the presented HCW system [1]. Alper [2] studied a benchmark building frame with and without bolted dissipative beam splices. The performance of both innovative and conventional structures has been quantified in terms of energy dissipation, floor displacements and inter-story drifts, as a result of nonlinear transient dynamic analysis. In the Morelli’s paper[3] the development, calibration and experimental validation of two component-based models of dissipative steel links connecting a reinforced concrete wall to a steel gravity frame is presented. Replaceable steel coupling beams (RSCB) [4] have been proposed as an alternative to conventional reinforced concrete (RC) coupling beams for enhanced seismic resiliency of coupled wall systems. This paper presents a series of quasi-static tests conducted to examine the seismic behaviour of RSCBs with RC slabs and to identify reasonable slab configurations that can minimize the damage to RC slabs. The performance of both innovative and conventional structures has been quantified in terms of energy dissipation, floor displacements and inter-story drifts, as a result of nonlinear transient dynamic analysis.

In these new frame type and the link we studied in the paper, damage concentrates mainly on the bolted dissipative beam splices (links) acting as “structural fuses”, which can be easily and inexpensively replaced after strong seismic events.

According to the above analysis, some plans and adjustments have been made for the follow-up works:

Firstly, we will continue to do some frame system tests and a lot of finite element analysis. We need to further advance the test method and the computational models to minimize the disparities as much as possible. After validating the accuracy of the finite element (FE) modelling approach against previous experiments, a comprehensive parametric computational study is conducted to investigate the main parameters and rules that affect the mechanical properties of structures.

Secondly, we need to do further theoretical analysis to establish the resilience model of the link.

Finally, the optimized design methodology based on the yielding criterion is proposed.

Sincerely thank you again for your valuable comments.

1.          AlessandroZona   Hervé Degée   Graziano Leoni. Ductile design of innovative steel and concrete hybrid coupled walls. J. Constr. Steel. Res. 2016117, 204-213.

2.          Alper, K. Milot, M.; Influence of repairable bolted dissipative beam splices (structural fuses) on reducing the seismic vulnerability of steel-concrete composite frames. Soil. Dyn. Earthq. Eng. 2019117, 281-298.

3.          Morelli, F. Manfredi, M. Salvatore, W.;  An enhanced component based model for steel connection in a hybrid coupled shear wall structure: development, calibration and experimental validation. Computers & Structures2016176, 50-69.

4.          Alireza, F. Farhad, K. Pouya, P.; Experimental study of a replaceable steel coupling beam with an end-plate connection. J. Constr. Steel. Res. 2016122, 138-150.

It is extremely annoying to review multiple times this manuscript and discover that the Authors have provided additional information and clarifications as a response to the Reviewer but have not included such information and clarifications in the text of their manuscript. 

Author Response

Dear reviewer

I am deeply sorry for the inconvenience caused to your work. In the last round of reply, due to the deviation in our understanding, we added the relevant content of our own test, and did not include other references into the manuscript. We apologize for this. Considering your suggestion, we made further revisions to our manuscript. The main corrections in the paper and the response to the reviewer’s comments are as follows:

Comment 1: The Authors have addressed most of the comments of the previous review round. However, it is not clear why the additional information provided in the Authors' response to report 2 have not been added in revised version of the manuscript.

Response 1: Thanks very much for your hard work. We have added related literature to the manuscript (line 43-54).New revisions are clearly highlighted in the paper.

1. Introduction

There are some researches have studied replaceable steel links in steel-concrete hybrid structures. AlessandroZona presented a hybrid coupled wall (HCW) system made of a single reinforced concrete (RC) wall coupled to two steel side columns by means of steel links. The design objective is to reduce or possibly avoid the damage in the RC wall while concentrating the seismic damage on the replaceable steel links [4]. Alper studied a benchmark building frame with and without bolted dissipative beam splices [5]. In the Morelli’s paper the development, calibration and experimental validation of two component-based models of dissipative steel links connecting a reinforced concrete wall to a steel gravity frame is presented [6]. Replaceable steel coupling beams (RSCB) [7] have been proposed as an alternative to conventional reinforced concrete (RC) coupling beams for enhanced seismic resiliency of coupled wall systems. In these new frame type, damage concentrates mainly on the bolted dissipative beam splices (links) acting as “structural fuses”, which can be easily and inexpensively replaced after strong seismic events.

References

4.        AlessandroZona   Hervé Degée   Graziano Leoni. Ductile design of innovative steel and concrete hybrid coupled walls. Constr. Steel. Res. 2016, 117, 204-213.

5.         Alper, K. Milot, M.; Influence of repairable bolted dissipative beam splices (structural fuses) on reducing the seismic vulnerability of steel-concrete composite frames. Soil. Dyn. Earthq. Eng. 2019, 117, 281-298.

6.         Morelli, F. Manfredi, M. Salvatore, W.; An enhanced component based model for steel connection in a hybrid coupled shear wall structure: development, calibration and experimental validation. Computers & Structures. 2016, 176, 50-69.

7.        Alireza, F. Farhad, K. Pouya, P.; Experimental study of a replaceable steel coupling beam with an end-plate connection. J. Constr. Steel. Res. 2016, 122, 138-150.

Sincerely thank you very much for all your help and look forward to hearing from you soon.

Best regards

This manuscript is a resubmission of an earlier submission. The following is a list of the peer review reports and author responses from that submission.